# Universal Semi-Supervised Learning

**Zhuo Huang**[1,2,3]**, Chao Xue**[3]**, Bo Han**[4]**, Jian Yang**[1,2]**, Chen Gong**[1,2,3](✉)
[1]PCA Lab, Key Lab of Intelligent Perception and Systems for
High-Dimensional Information of Ministry of Education
[2]Jiangsu Key Lab of Image and Video Understanding for Social Security,
School of Computer Science and Engineering, Nanjing University of Science and Technology
[3]JD Explore Academy [4]Department of Computer Science, Hong Kong Baptist University
{hzhuo, csjyang, chen.gong}@njust.edu.cn,
xuechao19@jd.com, bhanml@comp.hkbu.edu.hk

## Abstract

Universal Semi-Supervised Learning (UniSSL) aims to solve the open-set problem where both the class distribution (*i.e.*, class set) and feature distribution (*i.e.*, feature domain) are different between labeled dataset and unlabeled dataset. Such a problem seriously hinders the realistic landing of classical SSL. Different from the existing SSL methods targeting at the open-set problem that only study one certain scenario of class distribution mismatch and ignore the feature distribution mismatch, we consider a more general case where a mismatch exists in both class and feature distribution. In this case, we propose a "Class-shAring data detection and Feature Adaptation" (CAFA) framework which requires no prior knowledge of the class relationship between the labeled dataset and unlabeled dataset. Particularly, CAFA utilizes a novel scoring strategy to detect the data in the shared class set. Then, it conducts domain adaptation to fully exploit the value of the detected class-sharing data for better semi-supervised consistency training. Exhaustive experiments on several benchmark datasets show the effectiveness of our method in tackling open-set problems.

## 1   Introduction

A critical drawback of training a good neural network [25] is that it typically requires lots of labeled data, which is quite difficult to satisfy due to the unaffordable monetary cost as well as the huge demand for human resources. A popular way to solve such a problem is Semi-Supervised Learning (SSL) [7] which can effectively leverage scarce labeled data and abundant unlabeled data to train an accurate classifier. However, classical SSL [14, 15, 16, 17, 24, 39, 41, 49] relies on the *closed-set* assumption that the labeled and unlabeled data are drawn from the same class distribution and the same feature distribution. To be concrete, the class set of labeled data $\mathcal{C}^l$ is equal to that of unlabeled data $\mathcal{C}^u$, and the marginal distribution $p^l(\mathbf{x})$ of the labeled data is identical to the unlabeled feature distribution $p^u(\mathbf{x})$, which is shown in Figure 1 (a). However, in practice, the datasets at hand could greatly violate the above assumption by having both class distribution mismatch and feature distribution mismatch, which is denoted as *open-set*[1]. In this situation, traditional closed-set SSL would suffer from severe performance degradation [31].

Recent works [4, 9, 19, 22, 29, 31, 45] have focused on dealing with the class distribution mismatch problem to extend SSL into the wild. For example, Guo et al. (2020) [19] considers the *subset mismatch* situation when the classes in labeled data are only a subset of the classes in unlabeled data,

---

[1]Note that there are some other works [4, 45] also named as "open-set SSL". However, they all focus on the class distribution mismatch and do not deal with the feature distribution mismatch, so their setting is very different from ours.

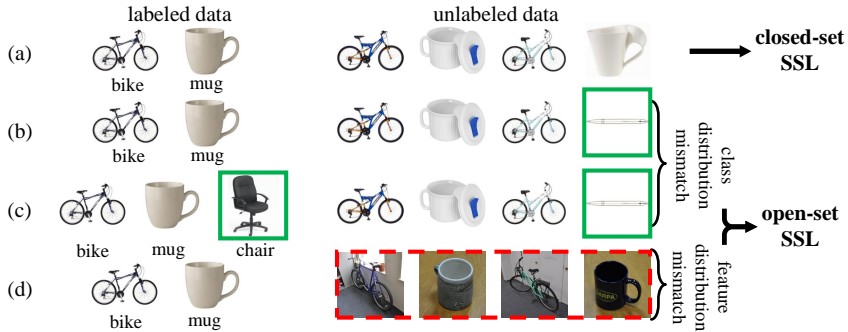

Figure 1: Problem illustration. (a) Closed-set SSL. (b) and (c) depict class distribution mismatch, where (b) describes subset mismatch; and (c) presents intersectional mismatch. (d) Feature distribution mismatch. In this figure, the dashed red box denotes feature distribution mismatch, and the solid green boxes mean the class distribution mismatch.

*i.e.*, $\mathcal{C}^l \subset \mathcal{C}^u$ and $\mathcal{C}^u \setminus \mathcal{C}^l \neq \varnothing$ (Figure 1 (b)). It utilizes a meta learning scheme to down-weight the unlabeled data that do not belong to $\mathcal{C}^l$. Chen et al. (2020) [9] deals with the *intersectional mismatch* problem where the labeled and unlabeled data both contain a shared class set but also hold a private class set, respectively, *i.e.*, $\mathcal{C}^l \cap \mathcal{C}^u \neq \varnothing$, $\mathcal{C}^l \setminus (\mathcal{C}^u \cap \mathcal{C}^l) \neq \varnothing$ and $\mathcal{C}^u \setminus (\mathcal{C}^u \cap \mathcal{C}^l) \neq \varnothing$ (Figure 1 (c)). It proposes a self-distillation method to filter out the probable unlabeled private data. However, existing methods require prior knowledge of the relationship between $\mathcal{C}^l$ and $\mathcal{C}^u$, which greatly limits their realistic application. When the class relationship is unknown, the potential private data from $\mathcal{D}^l$ and $\mathcal{D}^u$ could both seriously mislead the learning process. Moreover, existing works only consider the class distribution mismatch and totally ignore the feature distribution mismatch problem. The latter problem is also quite common in practice, as when we try to collect a large amount of unlabeled data to aid the model training, the feature distribution of the newly obtained unlabeled data could be heavily influenced by when, where, and how we collect them. As a result, the potential feature distribution difference between labeled and unlabeled data could seriously harm the learning performance. Therefore, it is necessary to design a universal method to solve different scenarios of class distribution mismatch and meanwhile deal with the feature distribution mismatch.

In this paper, we propose a new framework dubbed "Class-shAring data detection and Feature Adaptation (CAFA)" which is a Universal Semi-Supervised Learning (UniSSL) method for tackling different situations of open-set problem. Specifically, by considering that the labeled set and unlabeled set are drawn from different domains, we utilize a novel scoring mechanism to identify both the labeled and unlabeled data from the shared classes. The mechanism integrates two cues, namely domain similarity and label prediction shift, which are perfectly tailored for data detection in open-set SSL. Then we employ domain adaptation [5, 6, 12, 40, 43, 47] to match the identified unlabeled data to the same feature distribution of the class-sharing labeled data. After feature adaptation, the value of original unlabeled data can be fully exploited for boosting the learning performance. Moreover, we conduct weighted SSL to take full advantage of the class-sharing data from the open dataset. To sum up, our main contributions are:

- We propose a universal framework that can solve different scenarios of open-set SSL without any prior class knowledge.

- Our method can fully exploit the value of unlabeled data by mitigating the feature distribution mismatch between labeled data and unlabeled data.

- Experiments show that our method outperforms all other baselines in different situations of open-set problems.

## 2 Universal Semi-Supervised Learning

In our open-set SSL setting, we are given a labeled set $\mathcal{D}^l = \{(\mathbf{x}_i, y_i)\}_{i=1}^l$ containing $l$ instances $\mathbf{x}_i$ labeled with $\{y_i\}_{i=1}^l$, and an unlabeled set $\mathcal{D}^u = \{\mathbf{x}_j\}_{j=1}^u$ consisting of $u$ unlabeled instances $\mathbf{x}_j$, where $l \ll u$. The two datasets $\mathcal{D}^l$ and $\mathcal{D}^u$ are drawn from two different feature distributions $p^l(\mathbf{x}_i)$ and $p^u(\mathbf{x}_j)$, respectively. We use $\mathcal{C}^l$ to denote the labeled class set which contains the classes of labeled data, and employ $\mathcal{C}^u$ to represent the unlabeled class set consisting of the classes of unlabeled

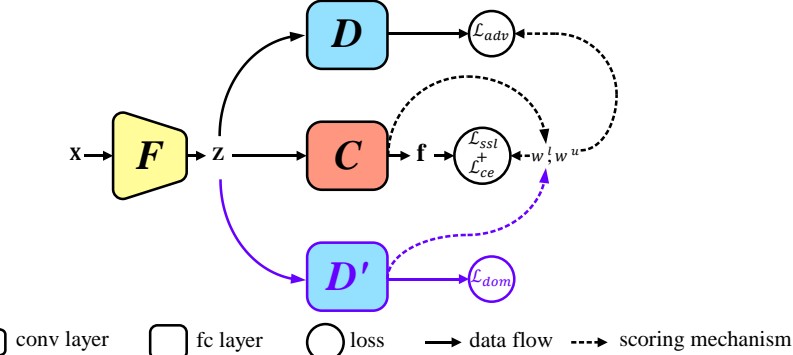

Figure 2: The pipeline of our Class-shAring data detection and Feature Adaptation (CAFA) approach.

data. Note that in our setting, we do not know the exact relationship between $\mathcal{C}^l$ and $\mathcal{C}^u$. Particularly, we use $\mathcal{C} = \mathcal{C}^l \cap \mathcal{C}^u$ to denote the common class set shared by $\mathcal{D}^l$ and $\mathcal{D}^u$, and use $\overline{\mathcal{C}}^l$ and $\overline{\mathcal{C}}^u$ to denote the class sets private to labeled data and unlabeled data, respectively. The feature distributions of labeled data with labels in $\mathcal{C}$ and $\overline{\mathcal{C}}^l$ are denoted as $p_{\mathcal{C}}^l(\mathbf{x}_i)$ and $p_{\overline{\mathcal{C}}^l}^l(\mathbf{x}_i)$, respectively, and the feature distributions of unlabeled data that belong to $\mathcal{C}$ and $\overline{\mathcal{C}}^u$ are denoted as $p_{\mathcal{C}}^u(\mathbf{x}_j)$ and $p_{\overline{\mathcal{C}}^u}^u(\mathbf{x}_j)$, respectively.

Our goal is to effectively identify the class-sharing data from both $\mathcal{D}^l$ and $\mathcal{D}^u$, and then eliminate the feature distribution mismatch between the identified labeled and unlabeled data to help train an accurate semi-supervised model in classifying the test data to the classes-of-interests $\mathcal{C}^l$.

## 2.1 The General Framework of CAFA

As shown in Figure 2, CAFA contains a feature extractor $F$, a classifier $C$, an adversarial discriminator $D$, and a non-adversarial discriminator $D'$. Given an input instance $\mathbf{x}$, we use $F$ to compute its feature representation $\mathbf{z} = F(\mathbf{x})$. Then we employ $C$ to output the label prediction $\mathbf{f}$ using $\mathbf{z}$. The non-adversarial discriminator $D'$ produces a domain similarity score $w_d$, which quantifies the similarity degree of an instance to one distribution. The adversarial discriminator $D$ aims to adversarially adapt the feature distributions of labeled and unlabeled data to the common classes $\mathcal{C}$.

The general framework of CAFA can be concisely formulated as:

$$
\min_{\theta_F, \theta_C} \max_{\theta_D} \underbrace{\mathbb{E}_{\mathbf{x}_i \sim p^l} \mathcal{L}_{ce}(C(F(\mathbf{x}_i)), y_i)}_{\text{supervised fidelity term}} - \gamma \underbrace{\mathbb{E}_{\mathbf{x}_i \sim p^l, \mathbf{x}_j \sim p^u} w^l \cdot w^u \cdot \mathcal{L}_{adv}(\mathbf{x}_i, \mathbf{x}_j; \theta_F, \theta_D)}_{\text{feature adaptation term}}
$$
$$
+ \delta \underbrace{\mathbb{E}_{\mathbf{x}_j \sim p^u} w^u \cdot \mathcal{L}_{ssl}(C(F(\mathbf{x}_j)), \mathbf{y}_j)}_{\text{class-sharing data exploration term}},
\tag{1}
$$

in which $\theta_F$, $\theta_C$, $\theta_D$ are the parameters of $F$, $C$, and $D$, respectively. In Eq. (1), the first term is dubbed *supervised fidelity term* which involves the standard cross-entropy loss $\mathcal{L}_{ce}(\cdot)$. The second term is dubbed *feature adaptation term* which introduces an adversarial learning loss $\mathcal{L}_{adv}$ to conduct feature adaptation on the class-sharing data from $\mathcal{D}^l$ and $\mathcal{D}^u$. Here the class-sharing data are detected through two scores $w^l$ and $w^u$ which will be detailed in Section 2.2. Through such a feature adaptation procedure, our CAFA approach can maximally exploit the unlabeled data and benefit SSL. The third term refers to *class-sharing data exploration term* which conducts semi-supervised training using a SSL loss $\mathcal{L}_{ssl}(\cdot)$ to make full use of class-sharing unlabeled data. Here the SSL loss can be any regularizer in existing methods such as consistency regularizer [24, 39] or manifold regularizer [1, 13, 44], and $\mathbf{y}_j$ is a $|\mathcal{C}^l|$-dimensional vector denoting the generated pseudo learning target for each unlabeled datum $\mathbf{x}_j$, where the notation $|\cdot|$ indicates the size of the corresponding set. The parameters $\gamma$ and $\delta$ are non-negative coefficients that trade off the above three terms.

From the general CAFA framework presented above, we can see that the class-sharing data detection, feature adaptation, and semi-supervised training are essential to our approach, and they will be detailed in Sections 2.2, 2.3, 2.4, respectively.

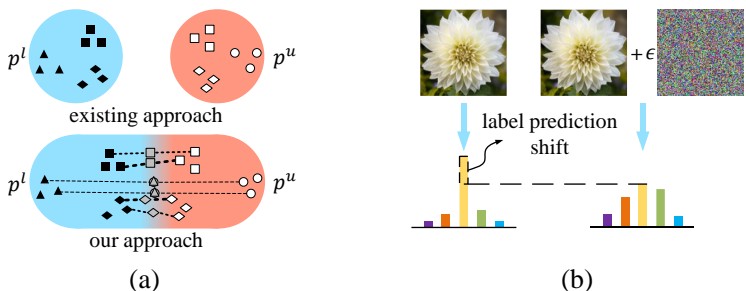

(a)

(b)

Figure 3: Illustration of our domain similarity and label prediction shift.

## 2.2 Class-Sharing Data Detection

Class-sharing data detection aims to correctly distinguish the training data belonging to $\mathcal{C}$ from those in $\overline{\mathcal{C}}^l \cup \overline{\mathcal{C}}^u$. To achieve this goal, we hope to model two class-sharing scores $w^l(\cdot)$ and $w^u(\cdot)$ for labeled and unlabeled data, respectively, which should satisfy the following inequalities [43]

$$
\begin{aligned}
\mathbb{E}_{\mathbf{x}\sim p_{\mathcal{C}}^l} w^l(\mathbf{x}) &> \mathbb{E}_{\mathbf{x}\sim p_{\overline{\mathcal{C}}^l}} w^l(\mathbf{x}), \\
\mathbb{E}_{\mathbf{x}\sim p_{\mathcal{C}}^u} w^u(\mathbf{x}) &> \mathbb{E}_{\mathbf{x}\sim p_{\overline{\mathcal{C}}^u}} w^u(\mathbf{x}).
\end{aligned}
\tag{2}
$$

The above inequities should hold in a large margin for better detection performance. Here we propose to utilize two cues, namely domain similarity $w_d$ and label prediction shift $w_s$, to model $w^l$ and $w^u$.

**Domain Similarity** has been employed to quantify whether an instance belongs to a specific domain by several methods [6, 43, 47]. They usually train a non-adversarial discriminator $D'$ to predict the data from $p^l$ as 0 and those from $p^u$ as 1 by minimizing a cross-entropy loss. The output value $w_d = D'(F(\mathbf{x}))$ can be considered as the domain similarity of the input $\mathbf{x}$. Particularly, the input $\mathbf{x}$ is likely to be sampled from $p^u$ if $w_d$ is large, and $p^l$ otherwise, formally

$$
\mathbb{E}_{\mathbf{x}\sim p_{\overline{\mathcal{C}}^l}} w_d(\mathbf{x}), \mathbb{E}_{\mathbf{x}\sim p_{\mathcal{C}}^l} w_d(\mathbf{x}) < \mathbb{E}_{\mathbf{x}\sim p_{\overline{\mathcal{C}}^u}} w_d(\mathbf{x}), \mathbb{E}_{\mathbf{x}\sim p_{\mathcal{C}}^u} w_d(\mathbf{x}).
\tag{3}
$$

However, such a training strategy lacks exploitation on the middle region between two feature distributions. Consequently, it is prone to overfit to the situation when $\mathbb{E}_{\mathbf{x}\sim p_{\overline{\mathcal{C}}^l}} w_d(\mathbf{x}) \approx \mathbb{E}_{\mathbf{x}\sim p_{\mathcal{C}}^l} w_d(\mathbf{x}) \approx 0$ and $\mathbb{E}_{\mathbf{x}\sim p_{\overline{\mathcal{C}}^u}} w_d(\mathbf{x}) \approx \mathbb{E}_{\mathbf{x}\sim p_{\mathcal{C}}^u} w_d(\mathbf{x}) \approx 1$, thus making the class-sharing data unrecognizable, as shown in the upper of Figure 3 (a). To solve this problem, we conduct Mixup [46] to strengthen the relationship between $\mathbf{x}_i \sim p_{\mathcal{C}}^l$ and $\mathbf{x}_j \sim p_{\mathcal{C}}^u$ to yield discriminative domain similarities. Specifically, given two feature representations $\mathbf{z}_i = F(\mathbf{x}_i)$ and $\mathbf{z}_j = F(\mathbf{x}_j)$ of labeled datum $\mathbf{x}_i$ and unlabeled datum $\mathbf{x}_j$, and their corresponding domain labels $d_i = 0$ and $d_j = 1$, respectively, we can generate a mixed feature representation $\tilde{\mathbf{z}}_{i,j}$ and a mixed domain label $\tilde{d}_{i,j}$ as following

$$
\tilde{\mathbf{z}}_{i,j} = \lambda \mathbf{z}_i + (1-\lambda)\mathbf{z}_j, \quad \tilde{d}_{i,j} = 1 - \lambda,
\tag{4}
$$

in which $\lambda$ is sampled from a Beta distribution $Beta(\alpha, \alpha)$ where $\alpha$ is a hyper-parameter. Then we leverage the mixed feature representations with their domain labels by adding an extra binary cross-entropy term to our domain similarity loss $\mathcal{L}_{dom}$, which is formulated as

$$
\begin{aligned}
\mathcal{L}_{dom} = &-\mathbb{E}_{\mathbf{x}_i\sim p^l} \log(1 - D'(F(\mathbf{x}_i))) - \mathbb{E}_{\mathbf{x}_j\sim p^u} \log D'(F(\mathbf{x}_j)) \\
&+ \mathbb{E}_{\mathbf{x}_i\sim p^l, \mathbf{x}_j\sim p^u} \left[ sim(\mathbf{z}_i, \mathbf{z}_j) \cdot (-(1-\lambda) \log D'(\tilde{\mathbf{z}}_{i,j}) - \lambda \log(1 - D'(\tilde{\mathbf{z}}_{i,j}))) \right],
\end{aligned}
\tag{5}
$$

where $sim(\mathbf{z}_i, \mathbf{z}_j) = \frac{\mathbf{z}_i^\top \mathbf{z}_j}{\|\mathbf{z}_i\|\|\mathbf{z}_j\|}$ denotes the cosine similarity between $\mathbf{z}_i$ and $\mathbf{z}_j$. Based on a reasonable assumption that $\mathbf{z}_i$ and $\mathbf{z}_j$ in $\mathcal{C}$ are closer to each other in the feature space than those in $\overline{\mathcal{C}}^l$ and $\overline{\mathcal{C}}^u$, such extra term weighted with cosine similarity can focus on interpolating the middle region between the two feature distributions $p_{\mathcal{C}}^l$ and $p_{\mathcal{C}}^u$, which helps preventing the aforementioned overfitting problem and making the domain similarity of class-sharing data closer to each other than the private data in $\overline{\mathcal{C}}^l$ and $\overline{\mathcal{C}}^u$, as shown in the lower panel of Figure 3 (a). Therefore, we can have $\mathbb{E}_{\mathbf{x}\sim p_{\overline{\mathcal{C}}^l}} w_d(\mathbf{x}) < \mathbb{E}_{\mathbf{x}\sim p_{\mathcal{C}}^l} w_d(\mathbf{x})$ and $\mathbb{E}_{\mathbf{x}\sim p_{\mathcal{C}}^u} w_d(\mathbf{x}) < \mathbb{E}_{\mathbf{x}\sim p_{\overline{\mathcal{C}}^u}} w_d(\mathbf{x})$. By combining Eq. (3), we have

$$
\mathbb{E}_{\mathbf{x}\sim p_{\overline{\mathcal{C}}^l}} w_d(\mathbf{x}) < \mathbb{E}_{\mathbf{x}\sim p_{\mathcal{C}}^l} w_d(\mathbf{x}) < \mathbb{E}_{\mathbf{x}\sim p_{\mathcal{C}}^u} w_d(\mathbf{x}) < \mathbb{E}_{\mathbf{x}\sim p_{\overline{\mathcal{C}}^u}} w_d(\mathbf{x}).
\tag{6}
$$

The above inequality would yield a larger margin than traditional training strategy when equipped with the Mixup training. Thereby, our domain similarity can better detect the class-sharing data in the open-set situation. Nevertheless, the domain similarity alone is not sufficient for class-sharing data detection. Therefore, we introduce label prediction shift to enhance the detection performance.

**Label Prediction Shift** indicates the influence imposed by adversarial perturbation on each instance, which can successfully differentiate the class-sharing data from private data, as shown in Figure 3 (b). Formally, given an input instance $\mathbf{x}$, its label prediction can be denoted as $\mathbf{f} = \left[ f_1(\mathbf{x}), f_2(\mathbf{x}), \cdots, f_{|\mathcal{C}^l|}(\mathbf{x}) \right]^\top$ where $\{f_i(\mathbf{x})\}_{i=1}^{|\mathcal{C}^l|}$ can be interpreted as the probability that $\mathbf{x}$ belongs to class $i$. Then we apply an adversarial perturbation on $\mathbf{x}$ to obtain the perturbed version of input instance $\mathbf{x}^* = \mathbf{x} - \epsilon \mathrm{sign}(-\nabla_{\mathbf{x}} \mathcal{L}_{ce}(\mathbf{x}, \max_{i \in \{1, \cdots, |\mathcal{C}^l|\}} f_i(\mathbf{x})))$, where $\epsilon$ controls the perturbation magnitude. As a result, the adversarial perturbation would decrease the maximum probability of the given input. Then, the label prediction shift can be computed as

$$w_s = \max_{i \in \mathcal{C}^l} f_i(\mathbf{x}) - \max_{i \in \mathcal{C}^l} f_i(\mathbf{x}^*). \tag{7}$$

As a result, the computed label prediction shift would satisfy the following inequality:

$$\mathbb{E}_{\mathbf{x} \sim p_{\overline{\mathcal{C}}^u}^u} w_s(\mathbf{x}) < \mathbb{E}_{\mathbf{x} \sim p_{\mathcal{C}}^u} w_s(\mathbf{x}) < \mathbb{E}_{\mathbf{x} \sim p_{\mathcal{C}}^l} w_s(\mathbf{x}) < \mathbb{E}_{\mathbf{x} \sim p_{\overline{\mathcal{C}}^l}^l} w_s(\mathbf{x}). \tag{8}$$

Intuitively, the learning on scarce labeled data is strongly dependent on the supervised cross-entropy loss $\mathcal{L}_{ce}$, while the unlabeled data that are learned with consistency regularization are much more robust against perturbations [24, 30, 39], thus it is natural to have $\mathbb{E}_{\mathbf{x} \sim p_{\overline{\mathcal{C}}^u}^u} w_s(\mathbf{x}), \mathbb{E}_{\mathbf{x} \sim p_{\mathcal{C}}^u} w_s(\mathbf{x}) < \mathbb{E}_{\mathbf{x} \sim p_{\overline{\mathcal{C}}^l}^l} w_s(\mathbf{x}), \mathbb{E}_{\mathbf{x} \sim p_{\mathcal{C}}^l} w_s(\mathbf{x})$. Moreover, the abundant unlabeled data have the effect of improving the model generalizability in SSL. In open-set situation, the model generalizability on the class set $\overline{\mathcal{C}}^l$ is greatly limited since there are only scarce labeled private data available, which makes such kind of data vulnerable against perturbations. On the contrary, the model learning on the classes in $\mathcal{C}$ is quite sufficient when compared with $\overline{\mathcal{C}}^l$ since there are both labeled and unlabeled data that can be leveraged. Hence, we have $\mathbb{E}_{\mathbf{x} \sim p_{\mathcal{C}}^l} w_s(\mathbf{x}) < \mathbb{E}_{\mathbf{x} \sim p_{\overline{\mathcal{C}}^l}^l} w_s(\mathbf{x})$. Additionally, the unlabeled private data in $\overline{\mathcal{C}}^u$ do not belong to any known classes and completely lie out of any known distribution. As mentioned in [27], the adversarial perturbation would have less influence on their maximal label predictions than those data in $\mathcal{C}^l$. Therefore, we can have $\mathbb{E}_{\mathbf{x} \sim p_{\overline{\mathcal{C}}^u}^u} w_s(\mathbf{x}) < \mathbb{E}_{\mathbf{x} \sim p_{\mathcal{C}}^u} w_s(\mathbf{x})$. Based on the above explanations, the inequality in Eq. (8) should hold.

To integrate the proposed two cues $w_d$ and $w_s$ based on Eqs. (6) and (8), we can compute $w^l$ and $w^u$ through

$$\begin{aligned} w^l(\mathbf{x}) &= w_d(\mathbf{x}) - w_s(\mathbf{x}), \quad \mathbf{x} \in \mathcal{D}^l, \\ w^u(\mathbf{x}) &= w_s(\mathbf{x}) - w_d(\mathbf{x}), \quad \mathbf{x} \in \mathcal{D}^u. \end{aligned} \tag{9}$$

Note that $w_d$ and $w_s$ are both normalized into interval $[0, 1]$ before computation. Through Eq. (9), our class-sharing scores can perfectly satisfy Eq. (2), hence they are effective for detecting the class-sharing data from both $\mathcal{D}^l$ and $\mathcal{D}^u$.

## 2.3 Feature Adaptation

After detecting the class-sharing data in the above subsection, now we should eliminate the feature distribution mismatch between $p_{\mathcal{C}}^l$ and $p_{\mathcal{C}}^u$ such that the value of unlabeled data can be properly extracted to aid the subsequent SSL. To this end, we treat the labeled data as target domain (*i.e.*, $d_i = 0, \mathbf{x}_i \in \mathcal{D}^l$) and the unlabeled data as the source domain (*i.e.*, $d_j = 1, \mathbf{x}_j \in \mathcal{D}^u$), and conduct adversarial domain adaptation [5, 6, 12, 40, 47] to achieve this goal. Particularly, we apply the class-sharing scores $w^l$ and $w^u$ to the adversarial learning loss $\mathcal{L}_{adv}$, and train the adversarial discriminator $D$ to distinguish the labeled and unlabeled data. Meanwhile, the feature extractor $F$ is trained to deceive $D$. The above adversarial process is formulated as the following min-max game:

$$\max_{\theta_F} \min_{\theta_D} \mathbb{E}_{\mathbf{x}_i \sim p^l, \mathbf{x}_j \sim p^u} \mathcal{L}_{adv}(\mathbf{x}_i, \mathbf{x}_j; \theta_F, \theta_D) = -\mathbb{E}_{\mathbf{x}_i \sim p^l} w^l(\mathbf{x}_i) \cdot \log(1 - D(F(\mathbf{x}_i)))$$
$$-\mathbb{E}_{\mathbf{x}_j \sim p^u} w^u(\mathbf{x}_j) \cdot \log D(F(\mathbf{x}_j)). \tag{10}$$

Thanks to the two class-sharing scores $w^l$ and $w^u$, we can successfully mitigate the feature distribution mismatch between $p_{\mathcal{C}}^l$ and $p_{\mathcal{C}}^u$ without being influenced by the irrelevant distributions $p_{\overline{\mathcal{C}}^l}^l$ and $p_{\overline{\mathcal{C}}^u}^u$. As we will show later in the experiments, our feature adaptation can re-discover the value of unlabeled data and boost the SSL performance.

### 2.4 Semi-Supervised Training

With the aforementioned class-sharing data detection and feature adaptation, we can take full advantage of the open-set information by alleviating the negative impact from both class distribution mismatch and feature distribution mismatch. Then, we should aim at effectively exploring the class-sharing unlabeled data, meanwhile weakening the negative impact from private data. Particularly, the private data in $\overline{\mathcal{C}}^l$ could mislead the unlabeled data transferring to the wrong classes, and the unlabeled private data in $\overline{\mathcal{C}}^u$ could be erroneously incorporated into network training, causing further performance degradation. To solve this problem, we propose the following SSL training strategy:

$$\min_{\theta_F, \theta_C} w^u(\mathbf{x}) \cdot \mathcal{L}_{ssl}(C(F(\mathbf{x})), \hat{\mathbf{y}}), \tag{11}$$

where $w^u(\mathbf{x})$ is employed to weaken the network learning on unlabeled private data, and $\hat{\mathbf{y}}$ indicates the calibrated pseudo target for each unlabeled datum to mitigate the misleading bias introduced by labeled private data. To calibrate the original biased pseudo target $\mathbf{y}$, we propose to utilize a weighted softmax function. Particularly, we compute the average weight of $w^l$ with respect to each class $c$ as

$$w_c^{avg} = \frac{1}{l} \sum_{i=1}^{l} \mathbb{I}(y_i = c) \cdot w^l(\mathbf{x}_i), \quad c \in \mathcal{C}^l. \tag{12}$$

Based on Eq. (2), the computed weight $w_c^{avg}$ would be large if $c$ is in $\mathcal{C}$, and be small if $c$ is in $\overline{\mathcal{C}}^l$. Then we can calibrate the pseudo target $\mathbf{y}$ through

$$[\hat{\mathbf{y}}]_c = \frac{w_c^{avg} \cdot \exp\left[\mathbf{y}\right]_c}{\sum_{i=1}^{|\mathcal{C}^l|} w_i^{avg} \cdot \exp\left[\mathbf{y}\right]_i}, \quad c \in \mathcal{C}^l, \tag{13}$$

where the notation $[\cdot]_c$ denotes the $c$-th entry of the input vector. Through such process, the entries of $\hat{\mathbf{y}}$ belonging to $\overline{\mathcal{C}}^l$ would be suppressed, and those belonging to $\mathcal{C}$ would be enhanced, which successfully alleviates the bias from the original target $\mathbf{y}$.

To sum up, our general framework can be instantiated by substituting Eq. (10) and Eq. (11) into the feature adaptation term and class-sharing data exploration term in Eq. (1). Later experiments will show that our CAFA framework can effectively tackle different scenarios of the open-set problem without any prior knowledge of the class relationship and achieve encouraging performances.

## 3 Related Work

### 3.1 Closed-Set SSL

Closed-set SSL deals with the problem when $\mathcal{C}^l = \mathcal{C}^u$ and $p^l(\mathbf{x}) = p^u(\mathbf{x})$. Early closed-set SSL methods such as Entropy Minimization [18] and Pseudo-Label (PL) [26] enforce the networks to make confident predictions on unlabeled data. Later, consistency-based methods such as $\Pi$-Model (PI) [24], Temporal Ensembling [24], and Mean Teacher (MT) [39] conduct consistency training between temporally or spatially different models. After that, Virtual Adversarial Training (VAT) [30] computes adversarial perturbations which maximally change the input image data to enhance the model robustness. Recent methods mostly rely on data augmentation to improve the network generalizability. For instance, MixMatch (MM) [3] and ReMixMatch [2] employ the Mixup [46] to augment the training data as well as the label information, which is beneficial to network training. FixMatch (FM) [38] utilizes the label predictions of weakly augmented image data to guide the learning of strongly augmented image data and achieves state-of-the-art performance. However, as mentioned before, closed-set SSL cannot perform satisfactorily in practice. Thereby, open-set SSL is designed to extend SSL into the wild.

### 3.2 Open-Set SSL

Open-set SSL is a new topic in SSL which aims to tackle the problem when $\mathcal{C}^l \neq \mathcal{C}^u$ and $p^l(\mathbf{x}) \neq p^u(\mathbf{x})$. Laine & Aila (2016) [24] and Oliver et al. (2018) [31] firstly raised the class distribution mismatch problem in open-set SSL, which is further investigated by many subsequent methods. For example, Uncertainty Aware Self-Distillation (UASD) [9] deals with the intersectional mismatch by proposing a self-distillation method to filter out the probable unlabeled private data. Then Safe Deep Semi-Supervised Learning (DS3L) [19] and Multi-Task Curriculum Framework (MTCF) [45] solve the subset mismatch by employing different weighting strategies to down-weight the unlabeled

private data. Recently, Cao et al. (2021) [4] uses contrastive learning [8, 42] to separate the unlabeled private data from the class-sharing data in the subset mismatch problem. Existing open-set SSL methods made some significant attempts toward a more practical setting. However, when the feature distribution mismatch between labeled and unlabeled data presents, almost all existing methods would fail.

### 3.3 Domain Adaptation

Domain adaptation aims to transfer knowledge from the labeled source domain to the unlabeled target domain, which is generally divided into four categories: 1) *closed-set domain adaptation* assumes that the class sets of the source and target data are the same. Existing methods [10, 11, 12, 28, 40, 48] aim to learn class discriminative and domain invariant features from source and target domains; 2) *partial domain adaptation* considers that the class set of source data is larger than that of target data. Some methods [5, 6, 47] aim to apply class-level weight on each source datum to achieve per-class distribution matching; 3) *open-set domain adaptation* denotes the situation when the target domain contains private classes. Busto et al. (2017) [32] trains SVMs to identify the target private classes. Saito et al. (2018) [37] employs an extra logit to the classifier to incorporate the unknown target classes. Zhuo et al. (2019) [50] proposes to leverage word vectors to recognize the open domains; and 4) *universal domain adaptation* requires no prior knowledge of the class relationship between source and target domains. You et al. (2020) [43] utilizes the domain knowledge and the entropy value to find the data in the shared classes. Saito et al. (2020) [36] employs neighborhood clustering [21] to help the feature alignment between the class-sharing source and target data.

## 4 Experiments

In this section, we first specify the implementation details[2] in Section 4.1. Then, to thoroughly validate the proposed CAFA approach, we conduct extensive experiments under different scenarios of open-set SSL by comparing our method with popular closed-set methods as well as several existing open-set methods in Section 4.2. Finally, we present the detailed performance analysis of our method in Section 4.3.

### 4.1 Experimental Setup

**Datasets.**  We use CIFAR-10 [23], Office-31 [35], and VisDA2017 [33] to evaluation our method. We denote the 10 classes from CIFAR-10 as "0" $\sim$ "9", the 31 classes from Office-31 as "0" $\sim$ "30", and the 12 classes from VisDA2017 as "0" $\sim$ "11". The numbers of labeled instances in CIFAR-10, Office-31, and VisDA2017 are set to 2,400, 100, and 1,800, respectively, and the number of unlabeled instances in CIFAR-10, Office-31, and VisDA2017 are set to 20,000, 400, and 20,000, respectively.

Firstly, we use CIFAR-10 to construct datasets with class distribution mismatch, which contains two scenarios: for subset mismatch, we choose the class from "0" $\sim$ "5" to form $\mathcal{C}^l$, and the classes from "0" $\sim$ "8" to form $\mathcal{C}^u$; and for intersectional mismatch, we choose the classes from "0" $\sim$ "5" to form $\mathcal{C}^l$, and the classes from "3" $\sim$ "8" to form $\mathcal{C}^u$. Then, we use Office-31 and VisDA2017 to create datasets with both class and feature distribution mismatch. Office-31 dataset contains three domains (A, D, W) and VisDA2017 contains two domains (Simulation, Reality). In the experiments, we choose the labeled data and unlabeled data from different domains to create a feature distribution mismatch. Particularly, in Office-31, we have six combinations of "labeled data domain/unlabeled data domain" including "A/D", "A/W", "D/A", "D/W", "W/A", and "W/D". For subset mismatch, we choose the classes from "0" $\sim$ "19" as $\mathcal{C}^l$, and the classes from "0" $\sim$ "29" as $\mathcal{C}^u$; and for intersectional mismatch, we choose the classes from "0" $\sim$ "19" as $\mathcal{C}^l$, and the classes from "9" $\sim$ "29" as $\mathcal{C}^u$. In VisDA2017, we choose the reality domain as labeled data domain, and simulation domain for unlabeled data domain. For subset mismatch, we choose the classes from "0" $\sim$ "8" as $\mathcal{C}^l$, and the classes from "0" $\sim$ "11" classes to form $\mathcal{C}^u$; and for intersectional mismatch, we choose the classes from "0" $\sim$ "8" as $\mathcal{C}^l$, and the classes from "3" $\sim$ "11" classes to form $\mathcal{C}^u$. For all experiments, we test on the dataset sampled from the same feature and class distribution as the labeled dataset.

**Compared Methods.**  The compared methods include some popular closed-set SSL methods: PI [24], PL [26], MT [39], VAT [30], MM [3], FM [38]; and several existing open-set SSL methods: UASD [9], DS3L [19], and MTCF [45]. Moreover, we train the network only using labeled data to form the "Supervised" baseline.

---

[2]More details can be found in the supplementary material.

Table 2: Averaged test accuracies (%) over three runs on Office-31 and VisDA2017 dataset with feature distribution mismatch. The best results are highlighted in **bold**. The notation "A/D" denotes that the labeled data are from A domain and unlabeled data are from D domain.

| Method | Office-31 | | | | | | VisDA2017 |
| | A/D | A/W | D/A | D/W | W/A | W/D | |
|---|---|---|---|---|---|---|---|
| Supervised | 57.07 | 58.89 | 58.23 | 62.89 | 52.96 | 54.48 | 78.29 |
| PI [24] | 49.29 | 57.99 | 75.71 | 71.83 | 68.74 | 55.94 | 27.00 |
| PL [26] | 52.97 | 58.59 | 33.59 | 52.89 | 34.32 | 43.64 | 18.40 |
| MT [39] | 69.34 | 70.49 | 55.65 | 65.19 | 54.40 | 65.34 | 20.12 |
| VAT [30] | 16.19 | 31.85 | 25.54 | 38.89 | 35.51 | 30.32 | 16.89 |
| MM [3] | 23.34 | 41.45 | 33.89 | 31.42 | 40.69 | 34.12 | 67.58 |
| FM [38] | 69.77 | 70.62 | 61.05 | 60.29 | 62.50 | 59.61 | 85.78 |
| UASD [9] | 54.29 | 65.99 | 63.09 | 66.69 | 43.20 | 50.32 | 47.22 |
| DS3L [19] | 55.97 | 47.28 | 53.26 | 51.08 | 36.95 | 52.71 | 60.28 |
| MTCF [45] | 38.99 | 42.93 | 46.19 | 36.95 | 40.76 | 47.28 | 56.08 |
| CAFA-PI (ours) | **81.97** | **83.57** | **79.04** | **76.44** | **74.59** | **80.48** | **91.02** |

**Implementation Details.** We implement all methods in PyTorch and run all experiments on a single Tesla V100 GPU. We use ResNet-50 [20] pre-trained on ImageNet [34] as the backbone network. The batch size is set to 100 for CIFAR-10 dataset and 64 for other datasets. We adopt SGD optimizer with the initial learning rate $3 \times 10^{-4}$. The perturbation magnitude $\epsilon$ is set to 0.014 and the Beta distribution parameter $\alpha$ is set to 0.75.

## 4.2 Open-Set Evaluation

In this subsection, we first evaluate our method on CIFAR-10 dataset under class distribution mismatch which includes subset mismatch and intersectional mismatch. Then we testify the effectiveness on Office-31 and VisDA2017 datasets under feature distribution mismatch. Finally, we consider a more complex situation when both class and feature distribution mismatch exists in the open-set to validate the capability of our method in achieving UniSSL.

### 4.2.1 Only Class Distribution Mismatch

To testify the effectiveness of our method under class distribution mismatch with no need for a prior relationship between $\mathcal{C}^l$ and $\mathcal{C}^u$, we create both subset mismatch and intersectional mismatch in CIFAR-10 dataset. The experimental results are shown in Table 1. We can see that our framework can largely enhance the simple PI method to achieve almost the best performances. Moreover, by using the strongest SSL method FM, our framework surpasses all other compared baselines with a large margin, which validates the ability of our method in handling both scenarios of class distribution mismatch.

Table 1: Averaged test accuracies (%) over three runs on CIFAR-10 with class distribution mismatch. The best results are highlighted in **bold**.

| Method | CIFAR-10 | |
| | Subset Mismatch | Intersectional Mismatch |
|---|---|---|
| Supervised | 76.13 | |
| PI [24] | 75.02 | 73.19 |
| PL [26] | 75.11 | 74.71 |
| MT [39] | 75.38 | 74.63 |
| VAT [30] | 76.07 | 75.25 |
| MM [3] | 79.08 | 78.43 |
| FM [38] | 80.19 | 80.01 |
| UASD [9] | 77.11 | 76.30 |
| DS3L [19] | 79.78 | 78.16 |
| MTCF [45] | 77.23 | 76.67 |
| CAFA-PI (ours) | 79.36 | 79.10 |
| CAFA-FM (ours) | **83.97** | **81.28** |

### 4.2.2 Only Feature Distribution Mismatch

Then, we consider the feature distribution mismatch problem where the labeled data and unlabeled data are sampled from different domains when $\mathcal{C}^l = \mathcal{C}^u$. The experimental results are shown in Table 2. We can see that in this situation, almost all closed-set and open-set methods show severe performance degradation than the supervised baseline. However, our method using PI as the backbone method outperforms all other methods as well as the supervised baseline, which certifies that by eliminating the feature distribution gap, our feature adaptation is beneficial for boosting the learning performance of SSL.

### 4.2.3 Both Class and Feature Distribution Mismatch

Finally, we conduct experiments under both feature distribution mismatch and class distribution mismatch. Here the latter also includes two scenarios: subset mismatch and intersectional mismatch. The experimental results are shown in Table 3. We can see that our method using PI again outperforms all other methods and the supervised baseline, and achieves encouraging results in both scenarios. Hence, it validates that the proposed CAFA framework can universally deal with unknown circumstances and tackle both mismatch problems to achieve encouraging results.

Table 3: Averaged test accuracies (%) over three runs on Office-31 and VisDA2017 dataset with feature distribution mismatch and class distribution mismatch. The best results are highlighted in **bold**. The notation "A/D" denotes that the labeled data are from A domain and unlabeled data are from D domain.

| Method | Subset Mismatch | | | | | | | Intersectional Mismatch | | | | | | |
|---|---|---|---|---|---|---|---|---|---|---|---|---|---|---|
| | Office-31 | | | | | | VisDA2017 | Office-31 | | | | | | VisDA2017 |
| | A/D | A/W | D/A | D/W | W/A | W/D | | A/D | A/W | D/A | D/W | W/A | W/D | |
| Supervised | 57.07 | 58.89 | 58.23 | 62.89 | 52.96 | 54.48 | 78.29 | 57.07 | 58.89 | 58.23 | 62.89 | 52.96 | 54.48 | 78.29 |
| PI [24] | 45.15 | 56.97 | 38.45 | 66.99 | 48.34 | 54.94 | 17.54 | 64.09 | 66.11 | 66.39 | 64.79 | 46.85 | 52.74 | 26.83 |
| PL [26] | 34.79 | 46.14 | 63.67 | 57.04 | 61.44 | 44.84 | 22.06 | 56.14 | 52.09 | 58.79 | 47.14 | 46.05 | 38.20 | 32.22 |
| MT [39] | 74.89 | 71.84 | 69.69 | 72.75 | 67.74 | 62.34 | 21.35 | 65.54 | 68.14 | 66.19 | 70.89 | 59.37 | 61.57 | 27.52 |
| VAT [30] | 26.19 | 28.89 | 49.89 | 57.24 | 49.36 | 41.14 | 35.56 | 23.64 | 27.50 | 40.04 | 43.54 | 23.45 | 32.66 | 19.67 |
| MM [3] | 53.80 | 57.06 | 54.34 | 49.45 | 61.41 | 55.97 | 70.32 | 59.78 | 59.23 | 62.50 | 61.41 | 55.97 | 47.82 | 66.34 |
| FM [38] | 68.74 | 69.34 | 60.64 | 52.88 | 63.39 | 55.62 | 83.17 | 66.99 | 64.12 | 62.19 | 65.44 | 57.93 | 55.76 | 85.57 |
| UASD [9] | 42.52 | 38.34 | 56.54 | 67.54 | 44.83 | 50.78 | 37.97 | 45.99 | 31.14 | 39.44 | 71.84 | 30.84 | 49.78 | 21.57 |
| DS3L [19] | 48.36 | 50.54 | 61.41 | 65.76 | 46.19 | 60.86 | 69.44 | 52.17 | 50.54 | 48.36 | 61.08 | 55.43 | 49.56 | 67.17 |
| MTCF [45] | 55.97 | 53.80 | 55.79 | 59.78 | 47.28 | 51.63 | 74.48 | 59.78 | 55.43 | 58.15 | 62.17 | 53.80 | 54.34 | 58.38 |
| CAFA-PI (ours) | **81.44** | **82.49** | **78.49** | **77.29** | **74.13** | **78.50** | **88.86** | **81.57** | **80.17** | **78.74** | **75.19** | **73.69** | **72.39** | **86.30** |

## 4.3 Performance Analysis

In this subsection, we conduct several analytical experiments using PI method as the backbone on the Office-31 dataset to evaluate our method.

**Class-Sharing Scores.** To testify the effectiveness of our class-sharing data detection, we plot the probability density curve of our class-sharing scores $w^l$ and $w^u$ in Figure 4, we can see that the score distribution is clearly separable between class-sharing data and private data, thus validating our hypothesis in Eq. (2).

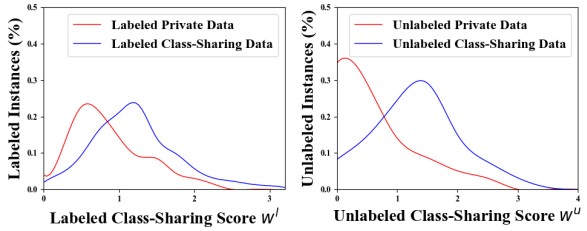

Figure 4: The probability density curve of $w^l$ and $w^u$.

**Averaged Weight.** To validate the performance of our pseudo target calibration, we plot the averaged weight $w_c^{avg}$ with respect to each class $c$ in Figure 5 (a). We can see that $w_c^{avg}$ is small when $c$ is in $\overline{\mathcal{C}}^l$, and large when $c$ belongs to $\mathcal{C}$. Therefore, our pseudo target calibration is helpful to mitigate the learning bias for unlabeled data.

**Robustness.** Finally, we vary the number of private classes from 0 to 10 to verify the robustness of our method against

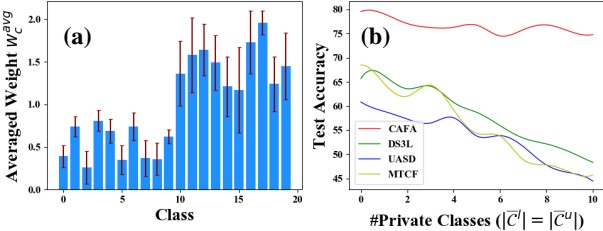

Figure 5: (a) The values of $w_c^{avg}$ over $\mathcal{C}^l$. We provide the average value over three runs with standard deviation. (b) The test accuracies with respect to different numbers of private classes.

different levels of class distribution mismatch. Here we set the number of labeled private classes $\overline{\mathcal{C}}^l$ equaling to unlabeled private classes $\overline{\mathcal{C}}^u$, hence the number of shared classes varies from 20 to 10. As shown in Figure 5 (b), we can see that our method surpasses all compared baselines with a large margin. Moreover, all compared methods show serious performance degradation when the number of private classes increases. However, our CAFA changes much flatter than other approaches, which indicates the robustness of our methods against various levels of class distribution mismatch.

## 5 Conclusion and Future Work

In this paper, we present a universal framework dubbed "Class-shAring data detection and Feature Adaptation" (CAFA) to solve different scenarios of open-set SSL problems with no need for any prior class knowledge. Particularly, we utilize a novel scoring mechanism that integrates the domain similarity and the label prediction shift to detect the data from the shared classes. Then we mitigate the feature distribution gap between the class-sharing data through domain adaptation to fully exploit the value of unlabeled data. Finally, we conduct semi-supervised training to properly learn consistent predictions between class-sharing labeled data and unlabeled data. Comprehensive experiments show the effectiveness of our CAFA framework on solving various open-set SSL problems. However, the proposed method is computationally expensive: during each round, our method requires an extra backward propagation to obtain the adversarial perturbation. Hence it is necessary to consider both effectiveness and efficiency in future work.

# 6 Acknowledgments and Disclosure of Funding

This research is supported by NSF of China (Nos: 61973162, U1713208), NSF of Jiangsu Province (No: BZ2021013), the Fundamental Research Funds for the Central Universities (Nos: 30920032202, 30921013114), CCF-Tencent Open Fund (No: RAGR20200101), Hong Kong Scholars Program (No: XJ2019036), RGC Early Career Scheme (No. 22200720), NSFC Young Scientists Fund (No. 62006202), HKBU CSD Departmental Incentive Grant, and "111" Program B13022.

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
