# Supplementary Material for Paper "Universal Semi-Supervised Learning"

In this supplementary material, we first provide the implementation details of our CAFA method and the baseline methods in Section A. Then we will explain the dataset establishment in Section B. Moreover, we will conduct additional experiments to further evaluate our method in Section C. Furthermore, we provide the standard deviation results that correspond to the main paper in Section D. Finally, we will discuss the limitations and social impact of our method in Section E.

## A  Implementation Details

To achieve a fair comparison, we investigate all algorithms with the same backbone network structure ResNet-50 [5]. In the CIFAR-10 dataset, we choose a batch size of 100, and in Office-31 and VisDA2017 datasets, we set the batch size to 64. For each iteration, the labeled data and unlabeled data are sampled in the same size, which equals half of batch size. The network training for all methods is iterated 10,000 times. We use an SGD optimizer with a weight decay factor of $5 \times 10^{-4}$ after 8,000 iterations. Other implementation details are presented below.

### A.1  Network Structure

Apart from the shared backbone network $F$, our method contains a classifier $C$ and two identical discriminators $D$ and $D'$. The classifier $C$ is one fully connected layer that maps the feature representations to label predictions. The discriminators $D$ and $D'$ have the same structure as [13], which is shown in Table 1. For the adversarial discriminator $D$, we implement the adversarial process using gradient reversal layer [3]. On the other hand, the training process of the non-adversarial discriminator $D'$ is detached from the backbone network $F$.

Table 1: Architecture of our discriminator $D$ and $D'$.

| Layer | Hyper-Parameters |
|---|---|
| GRL | flip-coefficient |
| Linear | 128→1,024 |
| ReLU | |
| Dropout | $p = 0.5$ |
| Linear | 1,024→1,024 |
| ReLU | |
| Dropout | $p = 0.5$ |
| Linear | 1,024→1 |
| Sigmoid | |

### A.2  Hyper-parameters

The trade-off parameters $\gamma$ and $\delta$ ramp up from 0 to 1 by following the functions $\gamma = \exp(-5 \times (1 - \min(\frac{iter}{8,000}, 1))^2)$ and $\delta = \exp(-5 \times (1 - \min(\frac{iter}{4,000}, 1))^2)$, respectively, where $iter$ denotes the current iteration. For other compared SSL methods, we follow [9] by using different hyper-parameter settings, which are shown in Table 2.

## B  Dataset Establishment

We have specified the classes that are chosen to construct $\mathcal{C}^l$ and $\mathcal{C}^u$ in the main paper, here we present other details for the establishment of our datasets with mismatched classes.

**CIFAR-10** is a typical dataset for SSL. It is composed of 50,000 training instances and 10,000 test instances collected from 10 natural categories. We use CIFAR-10 to create datasets with class distribution mismatch. For subset mismatch, there are 2,400 labeled training instances are chosen from $\mathcal{C}^l$, *i.e.*, 400 labeled instances per class. Then, we choose 20,000 unlabeled training instances from $\mathcal{C}^u$. Since $|\mathcal{C}^u|$ varies in different situations, there are approximately 2,222 unlabeled instances from each class for the subset mismatch and approximately 3,333 unlabeled instances for the intersectional mismatch. Furthermore, we select 5,000 test instances that belong to $\mathcal{C}^l$ to evaluate the performance of our method.

**Office-31** dataset is used to create a feature distribution mismatch by choosing the labeled and unlabeled data from different domains. It contains 3 domains: "Amazon" (A), "DSLR" (D), and "Webcam" (W), and each domain is composed of 31 classes. By changing the chosen domains, we have 6 combinations to form the labeled and unlabeled datasets: A/D, A/W, D/A, D/W, W/A, and W/D. For each combination, we choose 100 labeled instances from the classes in $\mathcal{C}^l$, *i.e.*, 5 labeled instances per class, and sample 400 unlabeled instances (we conduct up-sampling for some domains containing less than 400 instances) from the classes in $\mathcal{C}^u$, *i.e.*, 20 unlabeled instances per class for

Table 2: Hyperparameter settings of the compared SSL methods.

| Shared | |
|---|---|
| learning rate decay factor | 0.2 |
| # training iteration in which learning rate decay starts | 400,000 |
| # training iteration in which consistency coefficient ramp up starts | 200,000 |
| **Supervised** | |
| Initial learning rate | 0.003 |
| **Π-Model [6, 10]** | |
| Initial learning rate | $3 \times 10^{-4}$ |
| Max consistency coefficient | 20 |
| **Mean Teacher [12]** | |
| Initial learning rate | $4 \times 10^{-4}$ |
| Max consistency coefficient | 8 |
| Exponential moving average decay | 0.95 |
| **VAT [8]** | |
| Initial learning rate | 0.003 |
| Max consistency coefficient | 0.3 |
| $\epsilon$ | 6.0 |
| $\xi$ | $10^{-6}$ |
| **Pseudo-Label [7]** | |
| Initial learning rate | 0.003 |
| Max consistency coefficient | 0.3 |
| Pseudo-Label threshold | 0.95 |
| **MixMatch [1]** | |
| Initial learning rate | 0.003 |
| Augmentation number | 2 |
| Beta distribution $\alpha$ | 0.75 |
| **FixMatch [11]** | |
| Initial learning rate | 0.003 |
| Pseudo-Label threshold | 0.95 |
| **UASD [2]** | |
| Initial learning rate | 0.003 |
| Ensemble size | 10 |
| **DS3L [4]** | |
| Initial learning rate for backbone network | 0.003 |
| Initial learning rate for meta network | 0.001 |
| Initial learning rate for weighting network | $6 \times 10^{-5}$ |
| **MTCF [14]** | |
| Initial learning rate | 0.003 |

48    subset mismatch and approximately 13 unlabeled instances per class for the intersectional mismatch.
49    Moreover, we sample 500 test instances in $\mathcal{C}^l$ from the labeled domain to form a test set.

50    **VisDA2017** focuses on transferring knowledge from the simulated objects to real-world objects. It
51    contains a training dataset sampled from a simulation domain and a validation dataset sampled from
52    a reality domain, and each domain contains 12 classes. Here we choose 1,800 real-world instances
53    from the classes in $\mathcal{C}^l$ to form our labeled set, *i.e.*, 200 labeled instances per class. We choose 20,000
54    simulated instances from the classes in $\mathcal{C}^u$ to construct our unlabeled set, *i.e.*, approximately 1,667
55    unlabeled instances per class for the subset mismatch and 2,222 unlabeled instances per class for the
56    intersectional mismatch. Additionally, we choose 1,000 real-world instances in $\mathcal{C}^l$ to compose the
57    test set.

Table 3: Ablation studies on different model configurations. We report the averaged test accuracies±standard deviations (%) over three runs on Office-31 and VisDA2017 dataset with feature distribution mismatch and class distribution mismatch. The best results are highlighted in **bold**. The notation "W/A" denotes that the labeled data are from W domain and unlabeled data are from A domain.

| Method | Subset Mismatch | | Intersectional Mismatch | |
|---|---|---|---|---|
| | Office-31 (W/A) | VisDA2017 | Office-31 (W/A) | VisDA2017 |
| w/o Class-Sharing Data Detection | $65.76 \pm 8.49$ | $61.91 \pm 1.04$ | $70.65 \pm 4.53$ | $56.31 \pm 1.57$ |
| w/o Feature Adaptation | $56.84 \pm 5.92$ | $39.70 \pm 1.73$ | $51.41 \pm 7.13$ | $38.52 \pm 2.18$ |
| w/o Detection & Adaptation (PI) | $48.34 \pm 1.21$ | $17.54 \pm 10.5$ | $46.85 \pm 0.33$ | $26.83 \pm 12.3$ |
| CAFA-PI | $\mathbf{74.13 \pm 6.02}$ | $\mathbf{88.86 \pm 1.42}$ | $\mathbf{73.69 \pm 2.16}$ | $\mathbf{86.30 \pm 1.31}$ |

## C Ablation Study

In the main paper, we have provided the evaluation of the effectiveness of our method as well as some performance analyses. Here we conduct an ablation study to examine each module of the proposed CAFA framework, which includes *class-sharing data detection* and *feature adaptation*. Firstly, we conduct our framework without class-sharing data detection. Then we remove the feature adaptation module and conduct semi-supervised training on the detected class-sharing data. At last, we remove all the two modules and train the network using pure SSL. Here we use PI as the backbone method. The experimental results are shown in Table 3. We can see that all other model configurations have performance drops than the original CAFA framework. Hence both the two modules are essential for dealing with the open-set problems.

## D Evaluation Results with Standard Deviation

In this section, we present the accuracy±standard deviation of the evaluation results from the main paper, which are shown in Tables 4, 5, 6, and 7.

Table 4: Averaged test accuracies±standard deviations (%) over three runs on CIFAR-10 with class distribution mismatch. The best results are highlighted in **bold**.

| Method | CIFAR-10 | |
|---|---|---|
| | Subset Mismatch | Intersectional Mismatch |
| Supervised | $76.13 \pm 0.24$ | |
| PI [6] | $75.02 \pm 0.66$ | $73.19 \pm 0.59$ |
| PL [7] | $75.11 \pm 0.75$ | $74.71 \pm 0.39$ |
| MT [12] | $75.38 \pm 0.78$ | $74.63 \pm 0.50$ |
| VAT [8] | $76.07 \pm 0.84$ | $75.25 \pm 0.48$ |
| MM [1] | $79.08 \pm 1.20$ | $78.43 \pm 1.79$ |
| FM [11] | $80.19 \pm 0.55$ | $80.01 \pm 1.21$ |
| UASD [2] | $77.11 \pm 0.69$ | $76.30 \pm 0.91$ |
| DS3L [4] | $79.78 \pm 0.75$ | $78.16 \pm 0.78$ |
| MTCF [14] | $77.23 \pm 0.39$ | $76.67 \pm 0.96$ |
| CAFA-PI (ours) | $79.36 \pm 0.51$ | $79.10 \pm 0.72$ |
| CAFA-FM (ours) | $\mathbf{83.97 \pm 0.91}$ | $\mathbf{81.28 \pm 0.76}$ |

Table 5: Averaged test accuracies±standard deviations (%) over three runs on Office-31 and VisDA2017 dataset with feature distribution mismatch. The best results are highlighted in **bold**. The notation "A/D" denotes that the labeled data are from A domain and unlabeled data are from D domain.

| Method | Office-31 | | | | | | VisDA2017 |
|---|---|---|---|---|---|---|---|
| | A/D | A/W | D/A | D/W | W/A | W/D | |
| Supervised | $57.07 \pm 0.69$ | $58.89 \pm 0.48$ | $58.23 \pm 0.45$ | $62.89 \pm 0.72$ | $52.96 \pm 0.36$ | $54.48 \pm 0.58$ | $78.29 \pm 0.73$ |
| PI [6] | $49.29 \pm 1.40$ | $57.99 \pm 3.96$ | $75.71 \pm 3.67$ | $71.83 \pm 2.69$ | $68.74 \pm 2.62$ | $55.94 \pm 7.93$ | $27.00 \pm 6.27$ |
| PL [7] | $52.97 \pm 1.12$ | $58.59 \pm 1.23$ | $33.59 \pm 2.39$ | $52.89 \pm 1.59$ | $34.32 \pm 2.72$ | $43.64 \pm 2.30$ | $18.40 \pm 6.80$ |
| MT [12] | $69.34 \pm 1.84$ | $70.49 \pm 2.39$ | $55.65 \pm 7.46$ | $65.19 \pm 6.25$ | $54.40 \pm 10.1$ | $65.34 \pm 7.18$ | $20.12 \pm 8.77$ |
| VAT [8] | $16.19 \pm 8.97$ | $31.85 \pm 8.32$ | $25.54 \pm 8.34$ | $38.89 \pm 9.97$ | $35.51 \pm 8.08$ | $30.32 \pm 8.15$ | $16.89 \pm 3.74$ |
| MM [1] | $23.34 \pm 4.45$ | $41.45 \pm 9.85$ | $33.89 \pm 8.26$ | $31.42 \pm 8.22$ | $40.69 \pm 7.39$ | $34.12 \pm 6.12$ | $67.58 \pm 8.25$ |
| FM [11] | $69.77 \pm 1.33$ | $70.62 \pm 1.14$ | $61.05 \pm 3.65$ | $60.29 \pm 4.17$ | $62.50 \pm 2.73$ | $59.61 \pm 1.76$ | $85.78 \pm 2.85$ |
| UASD [2] | $54.29 \pm 2.34$ | $65.99 \pm 0.81$ | $63.09 \pm 1.58$ | $66.69 \pm 1.44$ | $43.20 \pm 1.97$ | $50.32 \pm 2.26$ | $47.22 \pm 2.67$ |
| DS3L [4] | $55.97 \pm 1.98$ | $47.28 \pm 1.38$ | $53.26 \pm 1.60$ | $51.08 \pm 2.29$ | $36.95 \pm 3.42$ | $52.71 \pm 2.31$ | $60.28 \pm 3.23$ |
| MTCF [14] | $38.99 \pm 3.01$ | $42.93 \pm 3.86$ | $46.19 \pm 2.68$ | $36.95 \pm 3.96$ | $40.76 \pm 3.98$ | $47.28 \pm 2.87$ | $56.08 \pm 3.75$ |
| CAFA-PI (ours) | $\mathbf{81.97 \pm 2.11}$ | $\mathbf{83.57 \pm 1.26}$ | $\mathbf{79.04 \pm 1.38}$ | $\mathbf{76.44 \pm 1.16}$ | $\mathbf{74.59 \pm 3.31}$ | $\mathbf{80.48 \pm 0.78}$ | $\mathbf{91.02 \pm 0.59}$ |

Table 6: Averaged test accuracies±standard deviations (%) over three runs on Office-31 and VisDA2017 dataset with feature distribution mismatch and subset class distribution mismatch. The best results are highlighted in **bold**. The notation "A/D" denotes that the labeled data are from A domain and unlabeled data are from D domain.

| Method | Office-31 | | | | | | VisDA2017 |
| | A/D | A/W | D/A | D/W | W/A | W/D | |
|---|---|---|---|---|---|---|---|
| Supervised | $57.07 \pm 0.69$ | $58.89 \pm 0.48$ | $58.23 \pm 0.45$ | $62.89 \pm 0.72$ | $52.96 \pm 0.36$ | $54.48 \pm 0.58$ | $78.29 \pm 0.73$ |
| PI [6] | $45.15 \pm 1.96$ | $56.97 \pm 2.67$ | $38.45 \pm 2.25$ | $66.99 \pm 4.53$ | $48.34 \pm 1.21$ | $54.94 \pm 6.92$ | $17.54 \pm 10.5$ |
| PL [7] | $34.79 \pm 6.16$ | $46.14 \pm 8.21$ | $63.67 \pm 9.61$ | $57.04 \pm 2.40$ | $61.44 \pm 5.97$ | $44.84 \pm 1.91$ | $22.06 \pm 6.09$ |
| MT [12] | $74.89 \pm 0.44$ | $71.84 \pm 0.58$ | $69.69 \pm 3.84$ | $72.75 \pm 12.3$ | $67.74 \pm 0.92$ | $62.34 \pm 1.06$ | $21.35 \pm 2.45$ |
| VAT [8] | $26.19 \pm 15.6$ | $28.89 \pm 7.56$ | $49.89 \pm 6.25$ | $57.24 \pm 3.42$ | $49.36 \pm 2.55$ | $41.14 \pm 11.7$ | $35.56 \pm 7.44$ |
| MM [1] | $53.80 \pm 3.67$ | $57.06 \pm 5.17$ | $54.34 \pm 1.64$ | $49.45 \pm 6.37$ | $61.41 \pm 4.06$ | $55.97 \pm 5.73$ | $70.32 \pm 1.00$ |
| FM [11] | $68.74 \pm 0.07$ | $69.34 \pm 1.15$ | $60.64 \pm 0.20$ | $52.88 \pm 2.16$ | $63.39 \pm 7.70$ | $55.62 \pm 6.67$ | $83.17 \pm 0.11$ |
| UASD [2] | $42.52 \pm 2.65$ | $38.34 \pm 0.29$ | $56.54 \pm 0.66$ | $67.54 \pm 0.93$ | $44.83 \pm 3.61$ | $50.78 \pm 1.82$ | $37.97 \pm 2.64$ |
| DS3L [4] | $48.36 \pm 1.29$ | $50.54 \pm 0.66$ | $61.41 \pm 5.90$ | $65.76 \pm 1.84$ | $46.19 \pm 4.72$ | $60.86 \pm 7.38$ | $69.44 \pm 1.85$ |
| MTCF [14] | $55.97 \pm 8.93$ | $53.80 \pm 4.90$ | $55.79 \pm 8.82$ | $59.78 \pm 8.12$ | $47.28 \pm 4.52$ | $51.63 \pm 3.83$ | $74.48 \pm 1.29$ |
| CAFA-PI (ours) | $\mathbf{81.44 \pm 2.89}$ | $\mathbf{82.49 \pm 0.36}$ | $\mathbf{78.49 \pm 1.10}$ | $\mathbf{77.29 \pm 0.36}$ | $\mathbf{74.13 \pm 6.02}$ | $\mathbf{78.50 \pm 3.76}$ | $\mathbf{88.86 \pm 1.42}$ |

Table 7: Averaged test accuracies±standard deviations (%) over three runs on Office-31 and VisDA2017 dataset with feature distribution mismatch and intersectional class distribution mismatch. The best results are highlighted in **bold**. The notation "A/D" denotes that the labeled data are from A domain and unlabeled data are from D domain.

| Method | Office-31 | | | | | | VisDA2017 |
| | A/D | A/W | D/A | D/W | W/A | W/D | |
|---|---|---|---|---|---|---|---|
| Supervised | $57.07 \pm 0.69$ | $58.89 \pm 0.48$ | $58.23 \pm 0.45$ | $62.89 \pm 0.72$ | $52.96 \pm 0.36$ | $54.48 \pm 0.58$ | $78.29 \pm 0.73$ |
| PI [6] | $64.09 \pm 2.89$ | $66.11 \pm 4.90$ | $66.39 \pm 2.49$ | $64.79 \pm 3.16$ | $46.85 \pm 0.33$ | $52.74 \pm 1.25$ | $26.83 \pm 12.3$ |
| PL [7] | $56.14 \pm 3.61$ | $52.09 \pm 5.82$ | $58.79 \pm 9.34$ | $47.14 \pm 6.00$ | $46.05 \pm 6.25$ | $38.20 \pm 1.56$ | $32.22 \pm 0.47$ |
| MT [12] | $65.54 \pm 1.96$ | $68.14 \pm 5.29$ | $66.19 \pm 1.69$ | $70.89 \pm 2.82$ | $59.37 \pm 0.53$ | $61.57 \pm 0.19$ | $27.52 \pm 7.78$ |
| VAT [8] | $23.64 \pm 8.90$ | $27.50 \pm 12.4$ | $40.04 \pm 5.98$ | $43.54 \pm 0.66$ | $23.45 \pm 4.73$ | $32.66 \pm 1.65$ | $19.67 \pm 2.03$ |
| MM [1] | $59.78 \pm 5.98$ | $59.23 \pm 5.70$ | $62.50 \pm 14.4$ | $61.41 \pm 9.34$ | $55.97 \pm 10.6$ | $47.82 \pm 6.65$ | $66.34 \pm 8.90$ |
| FM [11] | $66.99 \pm 1.21$ | $64.12 \pm 0.35$ | $62.19 \pm 0.21$ | $65.44 \pm 6.50$ | $57.93 \pm 0.10$ | $55.76 \pm 2.48$ | $85.57 \pm 0.96$ |
| UASD [2] | $45.99 \pm 5.12$ | $31.14 \pm 1.18$ | $39.44 \pm 4.09$ | $71.84 \pm 2.65$ | $30.84 \pm 1.15$ | $49.78 \pm 0.21$ | $21.57 \pm 4.16$ |
| DS3L [4] | $52.17 \pm 1.93$ | $50.54 \pm 2.65$ | $48.36 \pm 2.73$ | $61.08 \pm 1.83$ | $55.43 \pm 1.18$ | $49.56 \pm 0.73$ | $67.17 \pm 0.66$ |
| MTCF [14] | $59.78 \pm 4.42$ | $55.43 \pm 9.43$ | $58.15 \pm 2.12$ | $62.17 \pm 6.50$ | $53.80 \pm 5.20$ | $54.34 \pm 2.58$ | $58.38 \pm 0.82$ |
| CAFA-PI (ours) | $\mathbf{81.57 \pm 0.76}$ | $\mathbf{80.17 \pm 1.38}$ | $\mathbf{78.74 \pm 1.10}$ | $\mathbf{75.19 \pm 2.56}$ | $\mathbf{73.69 \pm 2.16}$ | $\mathbf{72.39 \pm 0.30}$ | $\mathbf{86.30 \pm 1.31}$ |

# E    Limitation and Social Impact

The proposed CAFA framework can solve different scenarios of open-set problems. However, there are still some limitations: 1) our method is computationally expensive. It requires an extra backward propagation to compute the adversarial perturbation; 2) the CAFA method ignores the potential class imbalance problem in the intersectional mismatch scenario. The number of instances in $\overline{\mathcal{C}}^l$ is much less than the instances in $\mathcal{C}^u$. As a result, such an imbalance problem could hurt the learning performance and it is worthy of further research; and 3) our method is a little bit complex since it needs to tackle different problems that occur in the open-set cases. Hence, it is necessary to design a more compact framework that can tackle both class and feature distribution mismatch.

Regardless of the limitations, our method could have some positive social impacts. As demonstrated in the introduction, our method is the closest method to reality than all other SSL approaches. Hence our method can be well conducted in many practical situations and help deploy SSL in modern industry.