# OpenReview forum: "Universal Semi-Supervised Learning"
_NeurIPS.cc/2021/Conference — NeurIPS 2021 Poster_

### Official Review · Reviewer_3d3x · 2021-07-03

**Rating:** 6
**Confidence:** 3

**Summary:**

The paper proposes an approach for universal semi-supervised setting that handles both class distribution and feature distribution mismatch. The key idea of the CAFA approach lies in introducing a scoring mechanism that identifies shared classes in the labeled and unlabeled data. The objective function in CAFA consists of supervised cross-entropy loss on labeled data, adversarial loss for feature adaptation between labeled and unlabeled data and class sharing data exploration term on pseudo-labels that are calibrated by weighted softmax function where weights are computed for each class. The approach is validated on three datasets CIFAR-10, Office-31 and VisDA on three tasks: (1) only class distribution mismatch,(2) only feature distribution mismatch, and (3) both class and feature distribution mismatch.

**Limitations And Societal Impact:**

Yes

**Main Review:**

Pros:
- The proposed universal SSL setting is very realistic and there is no method proposed specifically for this setting so far.
- The way scoring mechanism for labeled and unlabeled data is computed is interesting and provides insights about relationships of weights between shared and private classes of labeled and unlabeled data.
- Proposed approach significantly outperforms SSL methods.

Cons:
- The proposed approach is very complex and consists of so many design choices that are not clearly motivated and experimentally validated. Each design decision is supported only by an intuitive explanation why it should work.
- Why is L_ssl weighted by w_u when it does not consist of only private but also shared classes?
- What is the effect of pseudo-target calibration by w_c^avg?  While there is an experiment showing the average weight of classes in Fig 5, there is no ablation study that shows the effect on the performance.
- In domain similarity loss authors add an extra binary cross entropy term to help prevent overfitting but mixed feature representations will be generated from both domain-private and non-private classes so it is not clear what is the effect of this besides intuitive explanations that feature representations of shared classes should be closer.
- There are many parameters that are not explained how they should be scheduled and what is their effect on the performance. For example, why are \gamma and \delta in (1) scheduled as exp(-5(1-min(iter/n,1))^2)? Why is n 8000 for \gamma and 4000 for
\delta? How do different values affect performance? Similarly, how to schedule
 the parameter \lambda (i.e. alpha in Beta distribution)?
- How is the separation of the private classes in the unlabeled data guaranteed, or this can not be guaranteed at all and the approach can only guarantee to separate them from shared classes?  This should be clearly explained.
- Where is domain loss in eq (5) utilized? Eq (1) consists of L_{CE}, L_{adv} and L_{ssl}, but it is not clear where domain loss is used.
- Are the accuracies in Tables 1 and 3 reported only on the shared set of classes? If that is the case, does it mean that the authors report accuracy on only 3 classes for intersectional mismatch on the CIFAR-10?
- CIFAR-10 is quite a simple dataset, and it would be desirable to report performance on a more challenging CIFAR-100 dataset.
- Why do the authors consider 9 classes for CIFAR-10 instead of 10 and 30 classes instead of 31 on the Office dataset?
- In Figure 5b is this experiment done on CIFAR-10 dataset? Why is the performance of other baseline so low in the case when there are no private classes? How are methods evaluated when all classes are private?
- Lack of baselines. For feature distribution mismatch it is desirable to compare to the state-of-the-art partial and universal domain adaptation methods such as DANCE (Saito et al. Neurips ‘20) and ETN (Cao el al. CVPR ‘19)

Other comments:
- There is a mistake in the domain similarity paragraph. Non-adversarial discriminator D’ should predict data from p^l as 1 not 0.
- There is a typo in the number of examples of the Office dataset. I believe it should be 4000.


**Time Spent Reviewing:**

5

---

> ### Author Response · Authors · 2021-08-10
> **Official Comment for Reviewer 3d3x**
>
> R4:
> We thank Reviewer 3d3x for the detailed comments, and we have carefully addressed each of the concerns below.
>
> Re. the method is complex; the design choices are not clearly motivated and experimentally validated: Regarding the complexity of our method, the proposed framework mainly consists of three modules, namely class-sharing data detection, feature adaptation, and network training. The class-sharing data detection module aims to identify the class-sharing data by utilizing two cues: domain similarity output by a discriminator, and label prediction shift computed from adversarial perturbed inputs. feature adaptation and network training modules unify domain adaptation and SSL to properly exploit the unlabeled data. Through the above explanation, we see that our approach is actually very easy to follow.
>
> Our main goal is to tackle both feature distribution mismatch and class distribution mismatch, which are solved via using feature adaptation and class-sharing data detection. Particularly, our detection enabled by the proposed two cues can perfectly evaluate each example through domain information [D1] and prediction quality [D2] to help find the class-sharing data. To this end, we can solve the mismatch problems and activate SSL methods in the universal situation.
>
> For each module of our method, we have conducted ablation studies in Table 3 from the supplementary material. We can find that each module is indispensable for achieving improved results. Therefore, all design choices are well-motivated and experimentally validated.
>
> Re. the reason for applying $w_u$ on $L_{ssl}$: The main purpose of $w_u$ is to emphasize the shared classes and down-weight the private classes. As shown in Fig. 4, the generated $w_u$ of shared classes is larger than those of private classes. In such a way, we can retain the value of shared classes and leave out the harm from private data.
>
> Re. lack of ablation study on $w_c^{avg}$: To validate the effect of $w_c^{avg}$, we conduct an ablation study by removing the pseudo label calibration process. The experimental results of our method based on the backbone of PI model on Visda and Office (A/D) dataset are shown below:
>
> | dataset  | w/o calibration   | with calibration  |
> |:---:|:---:|:---:|
> | Visda    | 72.55  | 86.30  |
> | Office   | 67.79  | 81.57  |
>
> We can see that conducting pseudo label calibration is indeed beneficial to enhancing the learning performance.
>
>
> Re. the intuition and effect of the extra binary cross-entropy term in domain similarity loss: Since the class-sharing data belong to the same class set, their domain similarities would be closer to each other than the private data [D1], which would lead to larger class-sharing scores in Eq. (9). Through the feature adaptation weighted by such scores, the features of class-sharing data would also be closer to each other than the private data. As a sequel, although our Mixup training generates features from both private data and class-sharing data, it can still focus on leveraging the class-sharing data since their small feature distances yield larger cosine similarity values than the private data. Therefore, the proposed extra binary cross-entropy term can greatly enhance the class-sharing data detection.
>
>
> Re. parameter settings including the ramp-up schedules for $\gamma$, $\delta$ and $\alpha$ in Beta distribution: Since there are several relevant works that have the same parameter settings, therefore we follow [D3] to schedule our ramp-up function and follow [D4] to set $\alpha$ in the experiments. However, we still run some experiments on Visda dataset to analyze different parameter settings. ("*" denotes the parameter setting in our paper)
>
> | $\alpha$  | acc  | n in deciding $\gamma$  | acc  | n in deciding $\delta$  | acc  |
> |:---:|:---:|:---:|:---:|:---:|:---:|
> | 0.45  | 85.41   | 2000   | 85.10  | 2000   | 85.52 |
> | 0.6   | 85.86   | 4000   | 85.65  | 4000*   | 86.30 |
> | 0.75*  | 86.30   | 6000   | 85.32  | 6000   | 85.56 |
> | 0.9   | 84.97   | 8000*   | 86.30  | 8000   | 84.79 |
>
>
> Re. how to guarantee the separation of the private classes in the unlabeled data: As shown in Fig. 4, we can successfully separate the private classes from the shared classes with proper scores in both labeled dataset and unlabeled dataset. Since unlabeled data contain private classes and shared classes, the separation of unlabeled private data from unlabeled data can be guaranteed empirically.
>
> Re. the use of $L_{dom}$ in Eq (5): We would like to clarify that the $L_{adv}$ and $L_{dom}$ are the same things, and we will make the notations consistent in the final paper.
>
> Re. evaluation in Tables 1 and 3: We follow the same evaluation setting as [D5] and [D6] by employing a test set that shares the same classes with labeled data, i.e., 6 classes for CIFAR10, 20 classes for Office, and 9 classes for Visda. We have mentioned this in line 262 ("For all experiments, we test on the dataset sampled from the same feature and class distribution as the labeled dataset.").
>
> Re. performance on CIFAR-100 dataset: We conducted the experiments on CIFAR100 under the situation of intersectional mismatch. The results are reported below:
>
> |CAFA-PI (ours) | UASD-PI |  PI model  |  Fixmatch  |
> |:---:|:---:|:---:|:---:|
> | 58.53  |  56.33  | 38.39 | 56.34 |
>
> In the experiments, we use 0-59 classes as the labeled classes and 30-89 classes as the unlabeled ones. Although CIFAR100 is a much more challenging dataset than CIFAR10, our framework still shows effectiveness in tackling the intersectional mismatch problem.
>
>
> Re. numbers of classes chosen from CIFAR10 and Office datasets: Our setting of class distribution is identical to [D6] in which the mismatch proportion is set to 50%, except that we introduced more classes for constituting the unlabeled data. Specifically, [D6] uses 4 classes for unlabeled data in CIFAR10 while we use 6 classes. In Office dataset, the class distribution is similar to that in CIFAR10.
>
>
> Re. experimental setting and performance of SSL baselines in Figure 5b: 1) We would like to clarify that all the performance records are reported on Office dataset (line 309); 2) the reason why the baseline methods have poor performances is that they failed to solve the feature distribution mismatch problem; and 3) As for the situation when all classes of unlabeled data are private, it will against the basic setting of SSL as the unlabeled data become helpless in the learning process, hence we do not consider this situation.
>
> Re. lack of baselines such as DANCE & ETN: We want to emphasize that these domain adaptation methods are quite different from our setting for two reasons: 1) domain adaptation aims to transfer the knowledge from labeled source data to the target data to make correct predictions on the target data. On the contrary, universal SSL hopes to properly leverage the unlabeled data to boost the learning performance on labeled data by eliminating the distribution mismatch; 2) domain adaptation assumes that a large number of labels are available in the source domain, but SSL has only access to scarce labels. Based on these two reasons, domain adaptation methods like DANCE and ETN are not suitable for fair comparisons with SSL methods.
>
> Re. a mistake in the domain similarity paragraph and the typo in the number of examples of the Office dataset: Thanks for pointing out these typos. We will fix them in the final paper.
>
> We hope the responses could help address your concerns, and wish to receive your further feedback soon, thank you.
>
>
> [D1] Kaichao You, Mingsheng Long, Zhangjie Cao, Jianmin Wang, and Michael I. Jordan. Universal domain adaptation. In CVPR, 2020.
>
> [D2] Shiyu Liang, Yixuan Li, and Rayadurgam Srikant. Enhancing the reliability of out-of-distribution image detection in neural networks. In ICLR, 2018.
>
> [D3] Avital Oliver, Augustus Odena, Colin A Raffel, Ekin Dogus Cubuk, and Ian Goodfellow. Realistic evaluation of deep semi-supervised learning algorithms. In NeurIPS, 2018.
>
> [D4] David Berthelot, Nicholas Carlini, Ian Goodfellow, Nicolas Papernot, Avital Oliver, and Colin A Raffel. Mixmatch: A holistic approach to semi-supervised learning. In NeurIPS, 2019.
>
> [D5] Lan-Zhe Guo, Zhen-Yu Zhang, Yuan Jiang, Yu-Feng Li, and Zhi-Hua Zhou. Safe deep semi-supervised learning for unseen-class unlabeled data. In ICML, 2020.
>
> [D6] Yanbei Chen, Xiatian Zhu, Wei Li, and Shaogang Gong. Semi-supervised learning under class distribution mismatch. In AAAI, 2020.

---

> ### Author Response · Authors · 2021-08-24
> **Comment for Reviewer 3d3x**
>
> We hope that our reply to your comments could be helpful, and we are looking forward to hearing further feedback from you. Thank you very much!

---

### Official Review · Reviewer_A1eA · 2021-07-15

**Rating:** 9
**Confidence:** 4

**Summary:**

The paper investigates a realistic situation of open-set SSL, which includes the class distribution mismatch and the feature distribution mismatch. Moreover, it assumes the class relationship between labeled data and unlabeled data is unknown. The proposed method utilizes two scores to detect the data that belong to the common classes, and conduct domain adaptation to solve the feature mismatch problem.

**Limitations And Societal Impact:**

It seems that the proposed method is a bit complex. Besides, it would be better if the authors can discuss the relationship between the universal semi-supervised learning discussed in this paper and universal domain adaptation.
Other limitations have been indicated in the above weakness.

I think this work will attract intensive attention as this work studies a situation which is quite close to realistic applications. This work paves the way for applying semi-supervised learning to solving real-world problems.


**Main Review:**

The main contributions are: 1) to the best of my knowledge, this paper is the first work to investigate the domain difference between labeled and unlabeled data in SSL (in both label and feature domains), and it shows that conducting domain adaptation can address such problem; 2) the proposed domain similarity training which performs Mixup is novel; 3) this work shows that the adversarial perturbation can be utilized to perform OOD data detection, which is novel in open-set SSL.

Strengths:
- The problem setting is quite realistic, and the experimental results are also encouraging. I feel that this work will promote the application of semi-supervised learning to more real-world problems.
- The writing of this paper is good, and the details of algorithms are clearly stated.
- The proposed method is a general framework that can be deployed to many existing methods.


Weaknesses:
- The proposed method is a bit complex. Since it conducts both Mixup training and adversarial training in each round, the method could be computationally expensive.
- Can the proposed method outperform other SSL methods in the closed-set SSL setting?
- Is there any deeper analysis on why the closed-set SSL methods would fail in such a feature distribution mismatch setting?
- Since this work is strongly related to domain adaptation (the subset mismatch is related to partial domain adaptation, the intersectional mismatch is related to open-set domain adaptation, and the proposed setting is related to universal domain adaptation), it would be nice to discuss the difference between this problem and domain adaptation.


**Time Spent Reviewing:**

4

---

> ### Author Response · Authors · 2021-08-10
> **Official Comment for Reviewer A1eA**
>
> R3:
> We thank Reviewer A1eA for the positive comments.
>
> Re. the method is complex and computationally expensive: The proposed framework mainly contains three modules, namely class-sharing data detection, feature adaptation, and network training. The class-sharing data detection module aims to identify the class-sharing data by utilizing two cues: domain similarity output by a discriminator, and label prediction shift computed from adversarial perturbed inputs. feature adaptation and network training modules unify domain adaptation and SSL to properly exploit the unlabeled data. Through the above explanation, we see that our approach is actually very easy to follow.
>
> As for the computational complexity, our model is trained in a very efficient way. For example, our method only takes 12h to train a SSL classifier on CIFAR10 dataset (the number of examples is 22400) on a single Tesla V100 GPU with 16GB of memory.
>
>
> Re. performance on closed-set SSL setting: We conducted the experiments on CIFAR10 dataset in the closed-set setting, and the results are:
>
> | backbone method  | SSL  | CAFA  |
> |---|---|---|
> | PI model       | 79.10  | 80.34  |
> | Mean Teacher   | 80.54  | 81.33  |
>
> We can see that our framework achieves comparable performance with the traditional SSL methods in the closed-set situation. As mentioned in [C1] and [C2], closed-set SSL still suffers from a distribution gap between labeled and unlabeled data. However, our method can successfully eliminate such gap via using feature adaptation to slightly improve the performance.
>
>
> Re. deeper analysis on the failure of closed-set SSL methods in feature distribution mismatch: Traditional SSL performs well in closed-set because the label spaces of labeled data and unlabeled data are consistent, so utilizing unlabeled data can help the learning on labeled data. However, in the case of feature distribution mismatch, the biased unlabeled data distribution would mislead the learning on the labeled data, and thus causing the failure of closed-set SSL.
>
> Re. the relationship between universal SSL and different domain adaptation problems: We would like to emphasize the difference between SSL and domain adaptation, which are two folds: 1) domain adaptation aims to transfer the knowledge from labeled source data to the target data to make correct predictions on the target data. On the contrary, universal SSL hopes to properly leverage the unlabeled data to boost the learning performance on labeled data by eliminating the distribution mismatch; 2) domain adaptation assumes that a large number of labels are available in the source domain, but SSL has only access to scarce labels. Based on the above explanations, we see that the goal of our problem is very different from domain adaptation. In our method, we simply apply domain adaptation to close the gap between labeled and unlabeled data.
>
> We hope the responses could help address your concerns, and wish to receive your further feedback soon, thank you.
>
> [C1] Qin Wang, Wen Li, and Luc Van Gool. Semi-supervised learning by augmented distribution alignment. In ICCV, 2019.
>
> [C2] Christoph Mayer, Matthieu Paul, and Radu Timofte. Adversarial feature distribution alignment for semi-supervised learning. In AAAI, 2019.

---

### Official Review · Reviewer_3QaJ · 2021-07-16

**Rating:** 7
**Confidence:** 4

**Summary:**

The paper targets the universal semi-supervised learning problem where the class distribution and the feature distribution both mismatch between the labeled data and the unlabeled data.

The paper proposed a new framework Class-shAring data detection and Feature Adaptation (CAFA) to address the universal semi-supervised learning problem. The key insight is to design a scoring mechanism integrating domain similarity and label prediction shift to identify both the labeled and unlabeled data from the shared classes.

The paper conducts experiments on several datasets for semi-supervised learning and domain adaptation and shows competitive results on the semi-supervised learning setting with both class distribution mismatch and feature distribution mismatch.


**Limitations And Societal Impact:**

No obvious negative social impact.

**Main Review:**

Strength:

The problem setting tackled in this paper is quite practical and general for realistic applications. Since the unlabeled can be collected from any resource, there is potential that the unlabeled data have both different class distribution and feature distributions from the labeled data. Also, the problem setting is much more challenging than prior works with no prior knowledge of class distributions.

The paper addresses the overfitting problem of domain discriminators by training on mixup samples, which generates more discriminative domain similarities.

Label prediction shift with the adversarial perturbation can reflect the quality of the prediction.

The experiments are conducted on both semi-supervised learning dataset and domain adaptation datasets, which can demonstrate the effectiveness of the proposed method on the setting with feature distribution shift.

Weakness:

In Eqn. (1), the semi-supervised learning loss is performed on all the unlabeled data. As demonstrated in [14,25], if performing semi-supervised learning loss on all the unlabeled data, the performance degraded severely.

No variance is reported in the experiment results.

On line 109, both samples from p^u and p^l are labeled as '0'.

**Time Spent Reviewing:**

2

---

> ### Author Response · Authors · 2021-08-10
> **Official Comment for Reviewer 3QaJ**
>
>
> R2:
> We thank Reviewer 3QaJ for the positive comments.
>
> Re. the performance degradation caused by Eq. (1): In Eq. (1), we simply present the general SSL loss which does not explicitly reveal the fact that the class-sharing unlabeled data are weighted for training. In Eq. (11), this fact has been made clear.
>
> Re. variance in experimental results: In our experiments, each algorithm is implemented three times as in [B1], so the variance may not perfectly reflect the statistical characteristics. However, according to the reviewer’s comments, here we also present the accuracy±std on CIFAR10, Office (e.g., A/D denotes the labeled data are from A domain while unlabeled data are from D domain) and Visda datasets below:
>
> | Method         | Subset Mismatch | Intersectional Mismatch |
> |----------------|:---------------:|:-----------------------:|
> | Supervised     |    76.13±0.24   |        76.13±0.24       |
> | PI             |    75.02±0.66   |        73.19±0.59       |
> | PL             |    75.11±0.75   |        74.71±0.39       |
> | MT             |    75.38±0.78   |        74.63±0.50       |
> | VAT            |    76.07±0.84   |        75.25±0.48       |
> | MM             |    79.08±1.20   |        78.43±1.79       |
> | FM             |    80.19±0.55   |        80.01±1.21       |
> | UASD           |    77.11±0.69   |        76.30±0.91       |
> | DS3L           |    79.78±0.75   |        78.16±0.78       |
> | MTCF           |    77.23±0.39   |        76.67±0.96       |
> | CAFA-PI (ours) |    79.36±0.51   |        79.10±0.72       |
> | CAFA-FM (ours) |    83.97±0.91   |        81.28±0.76       |
>
> Feature Distribution Mismatch:
>
> | Method          | A/D          | A/W         | D/A         | D/W         | W/A         | W/D         | Visda       |
> |---|---|---|---|---|---|---|---|
> |Supervised       | 57.07±0.69   | 58.89±0.48  | 58.23±0.45  | 62.89±0.72  | 52.96±0.36  | 54.48±0.58  | 78.29±0.73  |
> |PI               | 49.29±1.40   | 57.99±3.96  | 75.71±3.67  | 71.83±2.69  | 68.74±2.62  | 55.94±7.93  | 27.00±6.27  |
> |PL               | 52.97±1.12   | 58.59±1.23  | 33.59±2.39  | 52.89±1.59  | 34.32±2.72  | 43.64±2.30  | 18.40±6.80  |
> |MT               | 69.34±1.84   | 70.49±2.39  | 55.65±7.46  | 65.19±6.25  | 54.40±10.1  | 65.34±7.18  | 20.12±8.77  |
> |VAT              | 16.19±8.97   | 31.85±8.32  | 25.54±8.34  | 38.89±9.97  | 35.51±8.08  | 30.32±8.15  | 16.89±3.74  |
> |MM               | 23.34±4.45   | 41.45±9.85  | 33.89±8.26  | 31.42±8.22  | 40.69±7.39  | 34.12±6.12  | 67.58±8.25  |
> |FM               | 69.77±1.33   | 70.62±1.14  | 61.05±3.65  | 60.29±4.17  | 62.50±2.73  | 59.61±1.76  | 85.78±2.85  |
> |UASD             | 54.29±2.34   | 65.99±0.81  | 63.09±1.58  | 66.69±1.44  | 43.20±1.97  | 50.32±2.26  | 47.22±2.67  |
> |DS3L             | 55.97±1.98   | 47.28±1.38  | 53.26±1.60  | 51.08±2.29  | 36.95±3.42  | 52.71±2.31  | 60.28±3.23  |
> |MTCF             | 38.99±3.01   | 42.93±3.86  | 46.19±2.68  | 36.95±3.96  | 40.76±3.98  | 47.28±2.87  | 56.08±3.75  |
> |CAFA-PI (ours)   | 81.97±2.11   | 83.57±1.26  | 79.04±1.38  | 76.44±1.16  | 74.59±3.31  | 80.48±0.78  | 91.02±0.59 |
>
> Subset Mismatch:
>
> | Method          | A/D          | A/W         | D/A         | D/W         | W/A         | W/D         | Visda       |
> |---|---|---|---|---|---|---|---|
> |Supervised       | 57.07±0.69   | 58.89±0.48  | 58.23±0.45  | 62.89±0.72  | 52.96±0.36  | 54.48±0.58  | 78.29±0.73  |
> |PI               | 45.15±1.96   | 56.97±2.67  | 38.45±2.25  | 66.99±4.53  | 48.34±1.21  | 54.94±6.92  | 17.54±10.5  |
> |PL               | 34.79±6.16   | 46.14±8.21  | 63.67±9.61  | 57.04±2.40  | 61.44±5.97  | 44.84±1.91  | 22.06±6.09  |
> |MT               | 74.89±0.44   | 71.84±0.58  | 69.69±3.84  | 72.75±12.3  | 67.74±0.92  | 62.34±1.06  | 21.35±2.45  |
> |VAT              | 26.19±15.6   | 28.89±7.56  | 49.89±6.25  | 57.24±3.42  | 49.36±2.55  | 41.14±11.7  | 35.56±7.44  |
> |MM               | 53.80±3.67   | 57.06±5.17  | 54.34±1.64  | 49.45±6.37  | 61.41±4.06  | 55.97±5.73  | 70.32±1.00  |
> |FM               | 68.74±0.07   | 69.34±1.15  | 60.64±0.20  | 52.88±2.16  | 63.39±7.70  | 55.62±6.67  | 83.17±0.11  |
> |UASD             | 42.52±2.65   | 38.34±0.29  | 56.54±0.66  | 67.54±0.93  | 44.83±3.61  | 50.78±1.82  | 37.97±2.64  |
> |DS3L             | 48.36±1.29   | 50.54±0.66  | 61.41±5.90  | 65.76±1.84  | 46.19±4.72  | 60.86±7.38  | 69.44±1.85  |
> |MTCF             | 55.97±8.93   | 53.80±4.90  | 55.79±8.82  | 59.78±8.12  | 47.28±4.52  | 51.63±3.83  | 74.48±1.29  |
> |CAFA-PI (ours)   | 81.44±2.89   | 82.49±0.36  | 78.49±1.10  | 77.29±0.36  | 74.13±6.02  | 78.50±3.76  | 88.86±1.42 |
>
> Intersectional Mismatch:
>
> | Method          | A/D          | A/W         | D/A         | D/W         | W/A         | W/D         | Visda       |
> |---|---|---|---|---|---|---|---|
> |Supervised       | 57.07±0.69   | 58.89±0.48  | 58.23±0.45  | 62.89±0.72  | 52.96±0.36  | 54.48±0.58  | 78.29±0.73  |
> |PI               | 64.09±2.89   | 66.11±4.90  | 66.39±2.49  | 64.79±3.16  | 46.85±0.33  | 52.74±1.25  | 26.83±12.3  |
> |PL               | 56.14±3.61   | 52.09±5.82  | 58.79±9.34  | 47.14±6.00  | 46.05±6.25  | 38.20±1.56  | 32.22±0.47  |
> |MT               | 65.54±1.96   | 68.14±5.29  | 66.19±1.69  | 70.89±2.82  | 59.37±0.53  | 61.57±0.19  | 27.52±7.78  |
> |VAT              | 23.64±8.90   | 27.50±12.4  | 40.04±5.98  | 43.54±0.66  | 23.45±4.73  | 32.66±1.65  | 19.67±2.03  |
> |MM               | 59.78±5.98   | 59.23±5.70  | 62.50±14.4  | 61.41±9.34  | 55.97±10.6  | 47.82±6.65  | 66.34±8.90  |
> |FM               | 66.99±1.21   | 64.12±0.35  | 62.19±0.21  | 65.44±6.50  | 57.93±0.10  | 55.76±2.48  | 85.57±0.96  |
> |UASD             | 45.99±5.12   | 31.14±1.18  | 39.44±4.09  | 71.84±2.65  | 30.84±1.15  | 49.78±0.21  | 21.57±4.16  |
> |DS3L             | 52.17±1.93   | 50.54±2.65  | 48.36±2.73  | 61.08±1.83  | 55.43±1.18  | 49.56±0.73  | 67.17±0.66  |
> |MTCF             | 59.78±4.42   | 55.43±9.43  | 58.15±2.12  | 62.17±6.50  | 53.80±5.20  | 54.34±2.58  | 58.38±0.82  |
> |CAFA-PI (ours)   | 81.57±0.76   | 80.17±1.38  | 78.74±1.10  | 75.19±2.56  | 73.69±2.16  | 72.39±0.30  | 86.30±1.31 |
>
> Re. line 109, both samples from p^u and p^l are labeled as '0': Sorry for the typo and we will fix it in the final paper.
>
> We hope the responses could help address your concerns, and wish to receive your further feedback soon, thank you.
>
> [B1] Kaichao You, Mingsheng Long, Zhangjie Cao, Jianmin Wang, and Michael I. Jordan. Universal domain adaptation. In CVPR, 2020.

---

> > ### Comment · Reviewer_3QaJ · 2021-08-16
> > **Response**
> >
> > Thanks for the response. That addresses my concerns. I will keep my score.

---

### Official Review · Reviewer_v3uq · 2021-07-19

**Rating:** 7
**Confidence:** 5

**Summary:**

This paper studies the universal semi-supervised learning, which combines traditional SSL and domain adaptation with the open-set problem. Under this setting, two issues are raised as the technical challenge, i.e., the class distribution shift and feature distribution shift. Accordingly, the CAFA is proposed in this paper, which identifies the class shared data among training and testing domain first, and then employs domain adaptation on these data for semi-supervised learning. Experiment results demonstrate the appilcability of the proposed method on several benchmark datasets.

**Ethical Concerns:**

There seems no ethical concern on this paper.

**Limitations And Societal Impact:**

See the detailed comment.

**Main Review:**

Originality:
This paper tries to unify SSL and domain adaptation in a single framework. The originality of is OK. But authors seem to miss an important reference which proposes the unsupervised open domain recognition problem:

[1] Unsupervised Open Domain Recognition by Semantic Discrepancy Minimization. In CVPR, 2019.

The key difference between CAFA and [1] lies in the usage of data that does not belong to the shared class. In CAFA, these data is assumed to be discarded, while in [1] these data are encouraged to be predicted to the categories (only in target domain) with equal probability. Authors should discuss the difference between CAFA and UODTN regarding to the problem setting.

Quality and clarity:
The quality and clarity of this paper is good.

Significance:
I think this paper has revealed a good point towards the development of SSL. However, the relation among SSL, unsupervised domain adaptation and other related problem setting (e.g., semi-supervised domain adaptation), should be discussed in an comprehensive way.

Detailed comments:

a. In CAFA, the authors choose to discard those data that does not belong to the shared class. This point still need to be justifed in a proper manner, since it has been proved that these private data can be used to enhance the prediction diversity as in [1] and [2].

[2] Towards Discriminability and Diversity: Batch Nuclear-norm Maximization under Label Insufficient Situations. CVPR, 2020.

Authors should compare CAFA with these methods on these methods under similar setting.

b. How does the wrongly detected class shared data affect the accuracy?

------
I have read the author response, which addressed my concerns appropriately. I stick to my original score and vote for acceptance on this paper.




**Time Spent Reviewing:**

5 hours

---

> ### Author Response · Authors · 2021-08-10
> **Official Comment for Reviewer v3uq**
>
> R1:
> We thank Reviewer v3uq for pointing out these two relevant works, and we will improve the completeness of the related work and the clarity of the problem setting.
>
> Re. the difference between CAFA (ours) and UODTN regarding problem setting: In UODTN, it assumes that the number of unknown classes is predefined. However, in CAFA, the number of unlabeled classes is entirely unknown. Therefore, it is impossible for us to classify the private classes correctly in our setting.
> Moreover, the universal SSL studied in this paper is quite different from domain adaptation. Specifically, we aim to mitigate the distribution mismatch between unlabeled data and labeled data such that the unlabeled data can be properly utilized to improve the learning performance on the known classes-of-interests from the labeled data distribution. On the contrary, the methods of domain adaptation try to transfer the knowledge from labeled source data to the target data so that the learned classifier can correctly predict the labels of target data.
>
> Re. justification on discarding the private data: In our Universal SSL, the private data are harmful to the training of classifier for identifying the classes-of-interests, as they are confusing for network training. Therefore, we should discard the detected private data, which is also the target of many existing works [A1], [A2]. Moreover, the two referred methods UODTN and BNM assume the number of private classes from the target domain is predefined. However, the exact number of unlabeled private classes is unknown in our problem setting. Thus, the diversity mentioned in UODTN and BNM is not applicable to our method.
>
> Re. comparison of CAFA with UODTN and BNM: For UODTN, it aims to solve unsupervised open domain recognition, which is very different from our SSL setting, so this method is not directly comparable with our CAFA. For BNM, we compare its instantiation on SSL with our CAFA on Visda dataset (number of shared class is 6), and the results below show the effectiveness of our method. The reason is that the SSL instantiation of BNM cannot deal with the feature mismatch problem.
>
> | CAFA-PI | BNM |
> | :--: | :--: |
> | 86.30 | 72.55 |
>
> Re. the impact of wrongly detected class: To understand the effect of wrongly detected class-sharing data, we manually set the class-sharing scores $w^u$ of the corresponding data to a small value (i.e., 0.3), and the generated accuracies on the Visda and Office datasets are 82.54% and 77.97% correspondingly. Therefore, the wrongly detected class-sharing data are harmful to the performance. Fortunately, our model can alleviate such negative cases as it can assign large values to the class-sharing data and small values to private data, as revealed by Fig. 5.
>
> We hope the responses could help address your concerns, and wish to receive your further feedback soon, thank you.
>
> [A1] Lan-Zhe Guo, Zhen-Yu Zhang, Yuan Jiang, Yu-Feng Li, and Zhi-Hua Zhou. Safe deep semi-supervised learning for unseen-class unlabeled data. In ICML, 2020.
>
> [A2] Yanbei Chen, Xiatian Zhu, Wei Li, and Shaogang Gong. Semi-supervised learning under class distribution mismatch. In AAAI, 2020.

---

### Decision · Program_Chairs · 2021-09-27

**Decision:**

Accept (Poster)

**Comment:**

This paper studies universal semi-supervised learning problem. In this setting, class distribution and feature distribution both mismatch between labeled and unlabeled data. The authors propose an algorithm CAFA and demonstrate its effectiveness on several benchmark data sets. The reviewers agree that the paper provide new insight on this problem and the setting is of practical importance. Good paper. I recommend accept.